# Identification of risk factors of Long COVID and predictive modeling in the RECOVER EHR cohorts
Chengxi Zang [1], Yu Hou[1], Edward J. Schenck[2], Zhenxing Xu[1], Yongkang Zhang[1], Jie Xu[3], Jiang Bian [3], Dmitry Morozyuk[1], Dhruv Khullar[1], Anna S. Nordvig[4], Elizabeth A. Shenkman[3], Russell L. Rothman[5], Jason P. Block[6], Kristin Lyman[7], Yiye Zhang[1], Jay Varma[1], Mark G. Weiner [1], Thomas W. Carton[7], Fei Wang [1] ✉ & Rainu Kaushal[1]

## Abstract

**Background** SARS-CoV-2-infected patients may develop new conditions in the period after the acute infection. These conditions, the post-acute sequelae of SARS-CoV-2 infection (PASC, or Long COVID), involve a diverse set of organ systems. Limited studies have investigated the predictability of Long COVID development and its associated risk factors. **Methods** In this retrospective cohort study, we used electronic healthcare records from two large-scale PCORnet clinical research networks, INSIGHT (~1.4 million patients from New York) and OneFlorida+ (~0.7 million patients from Florida), to identify factors associated with having Long COVID, and to develop machine learning-based models for predicting Long COVID development. Both SARS-CoV-2-infected and non-infected adults were analysed during the period of March 2020 to November 2021. Factors associated with Long COVID risk were identified by removing background associations and correcting for multiple tests. **Results** We observed complex association patterns between baseline factors and a variety of Long COVID conditions, and we highlight that severe acute SARS-CoV-2 infection, being underweight, and having baseline comorbidities (e.g., cancer and cirrhosis) are likely associated with increased risk of developing Long COVID. Several Long COVID conditions, e.g., dementia, malnutrition, chronic obstructive pulmonary disease, heart failure, PASC diagnosis U099, and acute kidney failure are well predicted (C-index > 0.8). Moderately predictable conditions include atelectasis, pulmonary embolism, diabetes, pulmonary fibrosis, and thromboembolic disease (C-index 0.7–0.8). Less predictable conditions include fatigue, anxiety, sleep disorders, and depression (C-index around 0.6). **Conclusions** This observational study suggests that association patterns between investigated factors and Long COVID are complex, and the predictability of different Long COVID conditions varies. However, machine learning-based predictive models can help in identifying patients who are at risk of developing a variety of Long COVID conditions.

## Plain language summary

Most people who develop COVID-19 make a full recovery, but some go on to develop post-acute sequelae of SARS-CoV-2 infection, commonly known as Long COVID. Up to now, we did not know why some people are affected by Long COVID whilst others are not. We conducted a study to identify risk factors for Long COVID and developed a mathematical modeling approach to predict those at risk. We find that Long COVID is associated with some factors such as experiencing severe acute COVID-19, being underweight, and having conditions including cancer or cirrhosis. Due to the wide variety of symptoms defined as Long COVID, it may be challenging to come up with a set of risk factors that can predict the whole spectrum of Long COVID. However, our approach could be used to predict a variety of Long COVID conditions.

The global COVID-19 pandemic starting in late 2019 has led to more than 557 million infections and 6.4 million deaths as of July 14, 2022[1]. Growing scientific and clinical evidence has demonstrated the existence of potential post-acute and long-term effects of COVID-19, which affect multiple organ systems[2] and are referred to as post-acute sequelae of SARS-CoV-2 infection (PASC, or Long COVID). Recently there have been several retrospective cohort analyses identifying potential PASC using real-world patient data[3–7]. However, research on the predictability of PASC and their associated risk factors is still limited, and mixed results have been reported. Such predictive modeling research can help patients and healthcare professionals recognize the risk of PASC early and inform effective actions. Several studies found older age, higher severities in the acute phase of SARS-CoV-2 infection[8], and

pre-existing conditions (e.g., hypertension, obesity) may be associated with a higher risk of developing PASC[9–14]. By contrast, some studies also reported that baseline clinical characteristics or demographics were not associated with PASC[12]. Two main challenges may explain these seemingly conflicting findings: 1) Prior studies have typically been conducted using patient cohorts with small sample sizes including only a few hundred or thousand patients[10,15], limiting the generalizability of conclusions derived, and 2) PASC conditions are highly heterogeneous concerning multi-organ manifestations[6,7], thus their predictabilities and associated risk factors could be heterogeneous as well.

To fill in the knowledge gap and address these challenges, we conducted a data-driven study on the predictability of a broad spectrum of incident PASC conditions and to identify their associated factors. We used two large electronic health records (EHR) cohorts from the PCORnet clinical research networks (CRN)[16], namely INSIGHT[17], covering patients in the New York City (NYC) area, and OneFlorida+[18], including patients from Florida. The INSIGHT and OneFlorida+ were used for primary analyses and validation respectively. A list of PASC conditions was selected based on our previous findings using a high-throughput data-driven analysis pipeline and existing evidence or clinical knowledge (See the method section for a detailed list of PASC diagnoses), which covered multiple organ systems[6,7,19]. Baseline covariates included basic demographics (e.g., age, gender, race, ethnicity), socioeconomic status, healthcare utilization history, body mass index, the period of infection, comorbidities, and the care settings in the acute phase including hospitalization and ICU stay. We used a regularized multivariate Cox proportional hazard model to uncover association maps between the abovementioned baseline covariates and different incident PASC conditions. Of note, the factors associated with PASC conditions were identified by removing background associations among non-infected patients and being selected with corrected significance levels due to multiple testings. We observed that severe acute SARS-CoV-2 infection, older age ($\geq 75$), female, extremes of weight, and having baseline comorbidities (e.g., cancer, chronic kidney disease, cirrhosis, coagulopathy, pregnancy, pulmonary circulation disorders) were associated with increased risk of a list of incident PASC patterns. Furthermore, we highlight that severe acute infections, being underweight, and having baseline conditions including cancer or cirrhosis are associated with having at least one PASC condition. We further developed machine learning-based prediction models to identify patients who were more likely to develop particular incident PASC conditions with their baseline characteristics and acute severity. We compared the performance of machine learning models with different levels of complexity, including regularized Cox proportional hazard model, regularized logistic regression, gradient boosting machine, and deep neural network in both the survival analysis setting and binary classification setting. We observed that it might be difficult to predict patients who will have at least one PASC condition (denoted as Any PASC) because a variety of PASC conditions were less predictable and were less associated with upfront disease severity. However, a range of PASC conditions were reliably predictable (e.g., dementia, myopathies, cerebral ischemia, COPD, heart failure, hypotension, malnutrition, acute kidney failure, and non-specific PASC diagnoses U099).

In all, complex association patterns and a lack of predictability of several PASC conditions may represent a challenge for managing heterogeneous PASC conditions. However, leveraging machine learning-based predictive models and EHR databases can help in identifying patients who are at risk of developing different incident PASC conditions. Among complex association patterns, we highlight severe acute infections, being underweight, and having baseline conditions including cancer or cirrhosis are likely associated with increased risk of having incident PASC in the post-acute phase, suggesting further investigation of the association between COVID-19 treatment in adults who are at high risk for severe COVID-19 and the risk of PASC beyond the acute phase. This study is part of the NIH Researching COVID to Enhance Recovery (RECOVER) Initiative, which seeks to understand, treat, and prevent the post-acute sequelae of SARS-CoV-2 infection (PASC).

## Methods

### Data

This study leveraged two large-scale de-identified electronic healthcare record warehouses from the INSIGHT Clinical Research Network (CRN)[17] and the OneFlorida+ CRN[18]. The INSIGHT CRN contained longitudinally linked data of approximately 12 million patient encounters at hospitals in the New York City metropolitan area, and the OneFlorida+ CRN contained the EHR data of nearly 15 million patients from Florida and selected cities in Georgia and Alabama. The use of the INSIGHT data was approved by the Institutional Review Board (IRB) of Weill Cornell Medicine following NIH protocol 21-10-95-380 with protocol title: Adult PCORnet-PASC Response to the Proposed Revised Milestones for the PASC EHR/ORWD Teams (RECOVER). The use of the OneFlorida+ data for this study was approved under the University of Florida IRB number IRB202001831. All EHRs used in this study were appropriately deidentified and thus no informed consent from patients was obtained.

### Definition of Long COVID

The current definition of Post-acute Sequelae of SARS-CoV-2 (PASC, or Long COVID) in the RECOVER protocols is "ongoing, relapsing, new symptoms, or other health effects occurring four or more weeks after the acute phase of SARS-CoV-2 infection"[7,20]. We examined a broad list of likely PASC conditions as outcomes, including depressive disorders, anxiety disorder, unspecified post-COVID-19 conditions encoded by the ICD-10 code U099 (in effect since October 2021 and we used ICD-10 code B948 before the implementation of U099)[21], fever, malaise and fatigue, dizziness, malnutrition, fluid disorders, diabetes mellitus, edema, hair loss, paresthesia, dermatitis, chronic obstructive pulmonary disease (COPD), atelectasis, pulmonary fibrosis, dyspnea, acute pharyngitis, acute bronchitis, dementia, myopathies, cerebral ischemia, encephalopathy, cognitive problems, sleep disorders, headache, muscle weakness, fibromyalgia, joint pain, acute kidney failure, cystitis, genitourinary problems, constipation, gastroparesis, abdominal pain, gastroesophageal reflux disease, heart failure, hypotension, pulmonary embolism, thromboembolism, abnormal heartbeat, chest pain, and anemia. We compiled this list based on both our previous study[6,7,19,22,23] and evidence from other literature[3,4,7]. An incident condition is defined in the SARS-CoV-2 infected patients who had the condition from 31 days to 180 days after the SARS-CoV-2 infection but did not have the condition three years to seven days before. See Supplementary Data 1 for the diagnostic code list.

### Eligibility criteria and study population

Our study included adult patients aged 20 years or older with at least one SARS-CoV-2 polymerase chain reaction (PCR) test or antigen laboratory test between March 1st, 2020, and November 30th, 2021. We further required at least one diagnosis code within three years to seven days before the index date (referred to as the baseline period), and at least one diagnosis code from day 31 to day 180 after the index date (referred to as the post-acute phase or follow-up period), to ensure that patients were connected to the healthcare system and were alive beyond the acute phase. We followed each patient from day 31 after his/her index date until the day of the first target outcome, documented death, the latest date of any documented records in the database, 180 days after the baseline, or the end of our observational window (December 31, 2021), whichever came first. We leveraged two exposure groups: a) the SARS-CoV-2 infected group, for the association study and predictive modeling, and b) the non-infected group, to rule out background associations that were not specific to COVID-19 infection. The infected group included patients with any positive SARS-CoV-2 PCR test or positive antigen laboratory test. The index date was defined as the date of the first documented positive PCR or antigen test for patients in the infected group. The non-infected group included patients whose SARS-CoV-2 PCR or Antigen tests were all negative throughout the entire study period and with no documented COVID-19-related diagnoses at any time. The index date for patients in the non-infected group was

**Fig. 1 | Patient selection from the INSIGHT and OneFlorida+ Clinical Research Networks, March 2020 to November 2021. a** the INSIGHT cohort, and (**b**) the OneForida+ cohort.

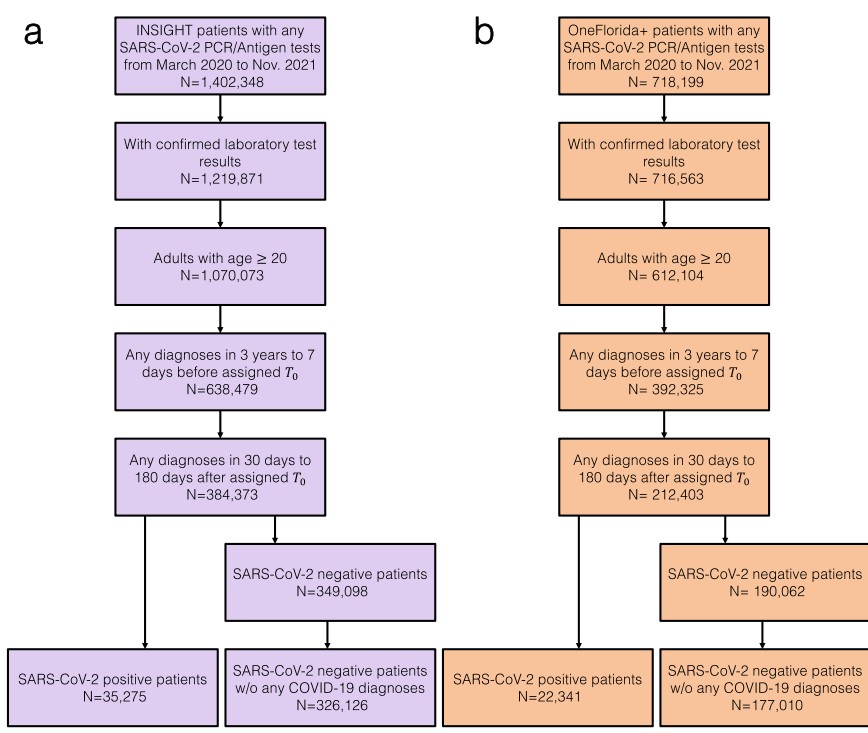

defined as the date of the first negative PCR or antigen test. The patient inclusion and exclusion cascades are illustrated in Fig. 1.

## Covariates

We built a list of 89 covariates that are potentially associated with PASC, including basic demographics (e.g., age, gender, race, ethnicity), socioeconomic status in terms of Area Deprivation Index (ADI)[21], healthcare utilization history, body mass index, the timing of infection, comorbidities, and the care settings in acute phase including hospitalization and ICU admission. For each of the categorical covariates, we defined its reference group the same as prior studies for acute SARS-CoV-2 infection[8]. We accounted for these baseline covariates by multivariate Cox proportional hazard model in our association analyses.

We collected clinical features in the baseline period (3 years to 1 week before lab-confirmed SARS-CoV-2 infection) and the severity of acute infection (1 week before to 2 weeks after lab-confirmed SARS-CoV-2 infection). Age was categorized into 20–39 years, 40–54 years, 55–64 years, 65–74 years, and 75+ years groups. We set 55–64 as the reference group. Gender was grouped into female and male (reference). Only three patients in INSIGHT were identified as other/missing gender who were excluded. The race was categorized into Asian, Black or African American, White (reference), other or missing. Ethnicity was grouped into Hispanic, not Hispanic (reference), and other/missing. We used the national-level ADI to capture the socioeconomic disadvantage of patients' residential neighborhoods[24]. Larger ADI values indicate mode socioeconomically deprived status. Missing ADI value was imputed with median ADI per site. The ADI is a ranking from 1 to 100 with 1 and 100 representing the lowest and the highest level of disadvantage, respectively. We grouped ADI into five categories and set the ADI category 1–20 as the reference group. Baseline healthcare utilization up to three years before the index date was measured according to their care setting. For each inpatient, outpatient, and emergency department encounter, we categorized each setting into 0 visits (reference group), 1 or 2 visits, and 3 or more visits, respectively. We also considered the infection time, which was grouped into March 2020–June 2020, July 2020–October 2020, November 2020–February 2021, March 2021–June 2021, and July 2021–November 2021. We set the first wave of the pandemic from March 2020 to June 2020 as the reference group. Of note, the third wave from July 2021 to November 2021 period was dominated by

the Delta variant. Body mass index (BMI) was grouped according to the WHO classification, BMI < 18.5 as underweight, BMI 18.5–24.9 as normal weight (reference), BMI 25–29.9 as overweight, BMI ≥ 30 obese, and set missing value as a separate group.

A wide range of baseline clinical comorbidities was collected, based on the Elixhauser comorbidities, conditions recommended by our clinician group, and related medications. Patients were ascertained as having a condition if they had at least two corresponding diagnoses documented during the baseline period[7]. We also counted the number of pre-existing conditions and grouped them into no comorbidity (reference)[1–5], or more.

## Association analysis

To uncover potentially complex association maps between baseline conditions and various incident PASC conditions, for each target PASC condition, we performed association analysis using the following two steps. In step I (multivariate association analysis for each PASC condition), we built a separate multivariate Cox proportional hazard model for each PASC condition in SARS-CoV-2 infected patients to assess associations of covariates and time to the first incident PASC event of interest event or censoring in the follow-up period (31-180 days after COVID-19 confirmation). The censoring event is defined as the earliest event of documented death, loss of follow-up in the database (the date of the last documented record in the EHR systems), 180 days after the baseline, or the end of our observational window (December 31, 2021). Fully adjusted hazard ratios (aHR) of each covariate and target PASC event were estimated. In step II (marginal associations due to SARS-CoV-2 infection), we built another multivariate Cox proportional hazard model for all the patients containing both SARS-CoV-2 infected and non-infected patients. The model inputs include two parts. One is the set of covariates. The other is the set of interaction terms defined as the product of each covariate and SARS-CoV-2 infection status (1 for SAR-CoV-2 infected patients and 0 for non-infected control patients) on the outcome condition. In this way, the coefficient of a particular covariate captured its association with the outcome condition for patients who were not infected by SARS-CoV-2, and the coefficient of its corresponding interaction term captured the "quantitative modifications" of such association for patients who were infected by SARS-CoV-2. Fully aHR of each covariate and interaction term was estimated on infected and non-infected combined patients.

A covariate was identified as a likely risk factor of a particular PASC condition if it satisfied the following four criteria: C1, The adjusted hazard ratio (aHR) estimated from the infected patients in Step I was greater than 1 when compared with the reference group; C2, The $p$-value of the above aHR by the Wald Chi-Square test was smaller than 0.000562, which was corrected by the Bonferroni method[25] for multiple testing; C3, the aHR of the interaction term of the corresponding covariate, namely the marginal increased risk due to SARS-CoV-2 infection, should be greater than 1 in Step II; and C4, the $p$-value of the aHR of the interaction term by the Wald Chi-Square test in the second Cox was smaller than 0.05. Of note, Criterion 3 and Criterion 4 try to identify the risk associations that have the portion that can be attributed to the SARS-CoV-2 infection, in addition to the background associations in non-infected patients. In summary, to uncover potentially complex association maps and to rule out background associations not specific to SARS-CoV-2 infection distinguish our method from existing association analysis literature.

## Machine learning-based predictive modeling

To uncover the predictability of different PASC conditions, we build predictive models for each PASC condition by examining machine learning models with varying complexity (including regularized Cox hazard model, regularized logistic regression, gradient boosting machine, and deep neural networks) in both survival analysis and binary classification settings.

For the survival analysis setting, we used a multivariate Cox proportional hazard model with L2 norm regularization to predict the time to the outcome event. For the binary classification setting, the occurrence of the target event in the follow-up period was labeled as 1 and 0 otherwise. We used logistic regression with L2 norm regularization, gradient boosting machine with random forest base learner, and deep feed-forward neural network. For each of the abovementioned models, the best model was selected by grid search (see details in the following sensitivity analysis paragraph) in a pre-defined hyper-parameter space through repeated cross-validation (ten times, five folds), detailed as follows: a) the regularized logistic regression, we adopted the L2-norm penalty and searched for the inverse of regularization strength from $10^{-3}$ to $10^3$ with 0.5 as the sampling step size; b) the gradient boosting machine with a random forest as the base learner[26], we searched hyperparameters from maximum depth (3,4,5), max number of leaves in one tree (10, 20, 30), and a minimal number of data in one leaf (30); c) deep forward neural network, we used the ReLU (Rectified Linear Unit) activation function for the hidden layer and searched the hidden layers ((32,), (64,), (128,), (32, 32), (64, 64), (128, 128)), and learning rate (0.001, 0.01, 0.1). For each of the above-mentioned models, the best model was selected by grid search of the corresponding hyperparameter space through repeated cross-validation (ten times, five folds). In the repeated cross-validation process, we set one of the folds as the test set and the rest of the data as the training set. The C-index and the area under the receiver operating characteristic curve (AUROC) were used to measure the predictive performance in the survival setting and binary classification setting, respectively.

The concordance index (C-index) and the area under the receiver operating characteristic curve (AUROC) were used to evaluate survival prediction performance and binary prediction performance respectively. Both two measures range from 0 to 1 with 0.5 indicating random guess and 1 indicating perfect prediction. The 95% confidence interval of the final performance was estimated by 1000-times bootstrapping performance on each of the testing datasets in repeated cross-validation.

## Stratified analysis

The stratified analysis was conducted by stratifying patients by their severity in the acute infection phase (hospitalized or non-hospitalized) and then performing statistical analysis within each stratum. The non-infected control patients were also stratified according to the hospitalized or non-hospitalized during the acute period (1 day before to 30 days after the index date), to capture background associations within each subgroup population.

## Sensitivity analysis

To get robust conclusions, we conducted the following sensitivity analyses. In addition to fully-adjusted association analysis, we also conduct univariate association analysis by using a univariate Cox model for each covariate. We also tested the impact of lifting Step II, namely without ruling out background associations not specific to SARS-CoV-2 infection, on the identified risk associations. On the other hand, we also investigate a shortened list of associations if we require the marginally increased risk to be significant. Specifically, we require the $p$-value of the Wald Chi-Square test of the interaction terms in the second Cox proportional hazard model <0.05. For the predictive modeling, we tested how different feature engineering methods will impact the predictive modeling. Rather than clinician-selected baseline predictors, we used a more high-dimensional feature engineering approach by using the first 3-characters of ICD-10 codes and medication at the ingredient level. These ICD-10 diagnosis codes and medications were selected to construct the input feature vectors of the prediction model based on the significant difference (P-value less than 0.0001 by Fisher's exact test) between patients with positive and negative PASC conditions results. After the feature selection process, the selected ICD-10 diagnosis codes, medication, and collected baseline covariates were constructed to represent every PASC condition.

## Validation analysis and generalizability

To get generalizable conclusions, we further replicated the abovementioned association analyses and predictive analyses in the OneFlorida+ cohort. The cohort selection and modeling strategies were the same as our primary analyses on the INSIGHT cohort.

## Reporting summary

Further information on research design is available in the Nature Portfolio Reporting Summary linked to this article.

## Results

### Factors associated with different PASC conditions

Here we analyzed association maps between baseline covariates and the risk of developing a range of incident PASC conditions. We quantified the association by three metrics including a) the unadjusted hazard ratio (HR), b) the fully adjusted hazard ratio (aHR), and c) the fully aHR filtered by significance corrected by multiple tests and positive marginal risks over the control group. We developed our primary results on the INSIGHT cohort (See the validation results on the OneFlorida+ cohort in the validation section) which included 35,275 adult patients with lab-confirmed SARS-CoV-2 infection and 326,126 non-infected control patients from March 2020 to November 2021 (see the inclusion-exclusion cascade in Fig. 1). Overall, among 35,275 enrolled SARS-CoV-2 infected patients in the INSIGHT cohort, 17,571 (49.8%) of them had at least one incident potential PASC condition (Table 1). The univariate HR and multivariate aHR between the covariates and the risk of getting at least one PASC condition were summarized in Table 2. Moreover, Fig. 2 summarizes fully adjusted aHRs that were significant under multiple tests ($p$-value < 0.000562 by the Wald Chi-Square test) and showed positive marginal risks over the control group. We summarize association results as follows.

### The severity of acute infection

Increased severity of the acute SARS-CoV-2 infection (according to the care settings) was associated with a higher risk of being diagnosed with incident conditions in the post-acute period. Overall, a higher risk of getting any incident diagnosis was observed in patients who were hospitalized during the acute phase (aHR 1.29 (1.24–1.33)) or in ICU (aHR 1.40 (1.32–1.49)) compared to patients who were not hospitalized during the acute phase (as a reference group, see Table 2 and Fig. 2). Specifically, as summarized in Fig. 2, severe acute infection was associated with a wide range of incident PASC conditions compared to non-hospitalized patients: the hospitalized patients showed higher risk of being diagnosed with sleep disorders (1.2-fold), chronic obstructive pulmonary disease (COPD, 1.7-fold), pulmonary fibrosis (2.1-fold), dyspnea (or shortness of breath, 1.8-fold), pulmonary

**Table 1 | Description of the lab-confirmed SARS-Cov-2 positive patients, with the number of at least one incident PASC diagnosis by patient characteristics, INSIGHT, March 2020–November 2021[a]**

| Characteristics | Number of patients with lab-confirmed SARS-CoV-2 infection (columns %) | Number of patients with lab-confirmed SARS-CoV-2 infection and with at least one incident PASC condition (columns %) |
|---|---|---|
| **Total** | 35,275 | 17,571 (49.8) |
| **The Severity of Acute Infection—no. (%)** | | |
| Not hospitalized | 22,148 (62.8) | 9809 (55.8) |
| Hospitalized w/o ICU | 11,480 (32.5) | 6611 (37.6) |
| ICU | 1647 (4.7) | 1151 (6.6) |
| **Age group—no. (%)** | | |
| 20- < 40 years | 9529 (27.0) | 3875 (22.1) |
| 40- < 55 years | 7975 (22.6) | 3850 (21.9) |
| 55- < 65 years | 6965 (19.7) | 3606 (20.5) |
| 65- < 75 years | 5712 (16.2) | 3170 (18.0) |
| 75+ years | 5094 (14.4) | 3070 (17.5) |
| **Sex—no. (%)** | | |
| Female | 20,686 (58.6) | 10,295 (58.6) |
| Male | 14,586 (41.3) | 7275 (41.4) |
| **Race—no. (%)** | | |
| Asian | 1736 (4.9) | 799 (4.5) |
| Black | 7791 (22.1) | 4029 (22.9) |
| White | 12,233 (34.7) | 5896 (33.6) |
| Other | 9844 (27.9) | 5208 (29.6) |
| Missing | 3671 (10.4) | 1639 (9.3) |
| **Ethnic group—no. (%)** | | |
| Hispanic | 10,658 (30.2) | 5789 (32.9) |
| Not Hispanic | 20,838 (59.1) | 10,305 (58.6) |
| Other/Missing | 3779 (10.7) | 1477 (8.4) |
| **Median area deprivation index (IQR)—rank** | | |
| ADI1-19 | 25,611 (72.6) | 12,715 (72.4) |
| ADI20-39 | 7891 (22.4) | 3962 (22.5) |
| ADI40-59 | 1126 (3.2) | 550 (3.1) |
| ADI60-79 | 162 (0.5) | 80 (0.5) |
| ADI80-100 | 485 (1.4) | 264 (1.5) |
| **Body mass index** | | |
| BMI: <18.5 underweight | 6419 (18.2) | 3685 (21.0) |
| BMI: 18.5- < 25 normal weight | 6431 (18.2) | 3272 (18.6) |
| BMI: 25- < 30 overweight | 8116 (23.0) | 4001 (22.8) |
| BMI: ≥30 obese | 9751 (27.6) | 4918 (28.0) |
| BMI: missing | 4558 (12.9) | 1695 (9.6) |
| **Index periods of patients—no. (%)** | | |
| 03/20–06/20 | 11,235 (31.8) | 5971 (34.0) |
| 07/20–10/20 | 2018 (5.7) | 1001 (5.7) |
| 11/20–02/21 | 14,637 (41.5) | 7256 (41.3) |
| 03/21–06/21 | 5573 (15.8) | 2802 (15.9) |
| 07/21–11/21 | 1812 (5.1) | 541 (3.1) |

**Table 1 (continued) | Description of the lab-confirmed SARS-Cov-2 positive patients, with the number of at least one incident PASC diagnosis by patient characteristics, INSIGHT, March 2020–November 2021[a]**

| Characteristics | Number of patients with lab-confirmed SARS-CoV-2 infection (columns %) | Number of patients with lab-confirmed SARS-CoV-2 infection and with at least one incident PASC condition (columns %) |
|---|---|---|
| **Pre-existing conditions—no. (%)[b]** | | |
| No comorbidity | 10,960 (31.1) | 4664 (26.5) |
| 1 comorbidity | 6277 (17.8) | 3012 (17.1) |
| 2 comorbidities | 3747 (10.6) | 1872 (10.7) |
| 3 comorbidities | 3008 (8.5) | 1571 (8.9) |
| 4 comorbidities | 2437 (6.9) | 1346 (7.7) |
| ≥5 comorbidities | 8846 (25.1) | 5106 (29.1) |
| Anemia | 4862 (13.8) | 2717 (15.5) |
| Arrhythmia | 5350 (15.2) | 3072 (17.5) |
| Asthma | 3950 (11.2) | 2179 (12.4) |
| Cancer | 3616 (10.3) | 2082 (11.8) |
| Chronic Kidney Disease | 5126 (14.5) | 2995 (17.0) |
| Chronic Pulmonary Disorders | 6209 (17.6) | 3511 (20.0) |
| Congestive Heart Failure | 3682 (10.4) | 2203 (12.5) |
| Coronary Artery Disease | 4658 (13.2) | 2652 (15.1) |
| Diabetes | 7681 (21.8) | 4310 (24.5) |
| Hypertension | 13,796 (39.1) | 7687 (43.7) |
| Mental Health Disorders | 3682 (10.4) | 2129 (12.1) |
| Prescription of Corticosteroids | 4999 (14.2) | 2695 (15.3) |
| Prescription of Immunosuppressant drug | 2110 (6.0) | 1086 (6.2) |

[a]The SARS-CoV-2-positive patients were identified by polymerase chain reaction (PCR) test or antigen test. The percentage may not sum up to 100 because of rounding. Category names are in bold.
[b]Coexisting conditions existed if two records in the 3 years before the index event. See all pre-existing conditions in Table 2.

embolism (1.4-fold), chest pain (1.3-fold), malnutrition (2.0-fold); while the ICU patients showed a higher risk of being diagnosed with myopathy (4.7-fold), cognitive problems (2.0-fold), anxiety disorder (1.8-fold), pulmonary fibrosis (2.1-fold), malnutrition (2.6-fold), malaise and fatigue (2.1-fold). In addition, concerning being diagnosed with general PASC codes U099/B948 (the U099 code, namely unspecified post-COVID-19 condition, was effective since 10/1/2021), hospitalized or ICU patients had a 2.2-fold and 4.3-fold higher risk respectively than non-hospitalized patients.

### Age
Patients aged 75 or older showed an increased risk of being diagnosed with a wide range of potential PASC conditions in the post-acute infection phase, including dementia (5.8-fold higher), cerebral ischemia (2.1-fold), malnutrition (1.8-fold) compared to patients were 55–64 years old (as reference).

### Gender and race
Female patients exhibited a 4.3-fold increased risk of being diagnosed with incident hair loss in the post-acute infection period compared to male

**Table 2 | Hazard ratios, 95% confidence intervals, and −log10(*P* values) for having at least one Post-acute Sequelae of SARS-CoV-2 Infection (PASC) condition, INSIGHT cohort, March 2020–November 2021[a]**

| Covariate | Univariate HR (95% CI) | Univariate −log10(*P* value) | Fully adjusted HR (95% CI) | Fully adjusted −log10(*P* value) |
|---|---|---|---|---|
| **The severity of acute infection** | | | | |
| Not hospitalized | 1.00 (ref) | | 1.00 (ref) | |
| Hospitalized | 1.48 (1.43–1.52) | 141.6 | 1.29 (1.24–1.33) | 46.4 |
| ICU | 1.46 (1.38–1.56) | 34.3 | 1.40 (1.32–1.49) | 25.9 |
| **Age group** | | | | |
| 20–39 years | 0.78 (0.75–0.81) | 31.8 | 0.88 (0.84–0.92) | 7 |
| 40–54 years | 0.93 (0.90–0.97) | 2.9 | 0.99 (0.95–1.03) | 0.2 |
| 55–64 years | 1.00 (ref) | | 1.00 (ref) | |
| 65–74 years | 1.12 (1.07–1.17) | 6.5 | 1.06 (1.01–1.11) | 1.8 |
| ≥75 years | 1.28 (1.22–1.34) | 25.9 | 1.14 (1.09–1.20) | 6.7 |
| **Sex** | | | | |
| Male | 1.00 (ref) | | 1.00 (ref) | |
| Female | 0.97 (0.94–1.00) | 1.4 | 1.04 (1.01–1.07) | 1.9 |
| **Race** | | | | |
| White | 1.00 (ref) | | 1.00 (ref) | |
| Asian | 0.91 (0.85–0.98) | 2 | 0.95 (0.89–1.02) | 0.7 |
| Black | 1.09 (1.05–1.13) | 4.9 | 1.05 (1.01–1.09) | 1.6 |
| Other | 1.11 (1.07–1.15) | 8.3 | 1.01 (0.97–1.05) | 0.2 |
| Missing | 0.90 (0.86–0.95) | 4 | 1.00 (0.95–1.06) | 0 |
| **Ethnic group** | | | | |
| Not Hispanic | 1.00 (ref) | | 1.00 (ref) | |
| Hispanic: Yes | 1.11 (1.08–1.15) | 10.8 | 1.09 (1.05–1.14) | 4.4 |
| Other or missing | 0.76 (0.73–0.81) | 23.4 | 0.86 (0.81–0.91) | 6.9 |
| **No. of hospital visits in the past 3 yrs** | | | | |
| Inpatient 0 | 1.00 (ref) | | 1.00 (ref) | |
| Inpatient 1–2 | 1.16 (1.12–1.20) | 15.3 | 0.96 (0.92–1.00) | 1.3 |
| Inpatient ≥3 | 1.17 (1.11–1.23) | 8.6 | 0.87 (0.82–0.93) | 4.1 |
| Outpatient 0 | 1.00 (ref) | | 1.00 (ref) | |
| Outpatient 1–2 | 0.71 (0.66–0.76) | 20.3 | 0.80 (0.74–0.86) | 9 |
| Outpatient ≥3 | 0.62 (0.58–0.66) | 57.7 | 0.64 (0.60–0.68) | 47.1 |
| Emergency 0 | 1.00 (ref) | | 1.00 (ref) | |
| Emergency 1–2 | 1.17 (1.13–1.21) | 19.3 | 1.05 (1.01–1.08) | 1.9 |
| Emergency ≥3 | 1.17 (1.12–1.22) | 13.6 | 1.02 (0.97–1.07) | 0.4 |
| **Median area deprivation index (rank)** | | | | |
| ADI1-19 (least deprived) | 1.00 (ref) | | 1.00 (ref) | |
| ADI20-39 | 0.99 (0.96–1.03) | 0.2 | 0.97 (0.93–1.00) | 1.2 |
| ADI40-59 | 0.97 (0.89–1.05) | 0.4 | 0.96 (0.89–1.05) | 0.4 |
| ADI80-100 | 1.09 (0.96–1.22) | 0.8 | 0.98 (0.87–1.10) | 0.1 |
| **Body mass index (kg/m²)** | | | | |
| BMI: 18.5- < 25 normal | 1.00 (ref) | | 1.00 (ref) | |
| BMI: <18.5 underweight | 1.18 (1.13–1.24) | 13.3 | 1.17 (1.12–1.23) | 10.8 |
| BMI: 25- < 30 overweight | 0.97 (0.93–1.01) | 0.9 | 0.97 (0.93–1.01) | 0.8 |
| BMI: ≥30 obese | 0.98 (0.94–1.02) | 0.4 | 0.98 (0.93–1.02) | 0.6 |
| BMI: missing | 0.69 (0.65–0.73) | 40.4 | 0.82 (0.77–0.87) | 11.1 |
| **Infection period** | | | | |
| 03/20–06/20 | 1.00 (ref) | | 1.00 (ref) | |
| 07/20–10/20 | 0.95 (0.89–1.01) | 1.1 | 0.98 (0.92–1.05) | 0.2 |
| 11/20–02/21 | 0.95 (0.92–0.98) | 3.1 | 0.99 (0.96–1.03) | 0.1 |
| 03/21–06/21 | 1.07 (1.02–1.11) | 2.5 | 1.07 (1.02–1.12) | 2.5 |
| 07/21–11/21 | 1.42 (1.30–1.54) | 14.8 | 1.55 (1.42–1.69) | 21.5 |

**Table 2 (continued) | Hazard ratios, 95% confidence intervals, and −log10(P values) for having at least one Post-acute Sequelae of SARS-CoV-2 Infection (PASC) condition, INSIGHT cohort, March 2020–November 2021ᵃ**

| Covariate | Univariate HR (95% CI) | Univariate −log10(P value) | Fully adjusted HR (95% CI) | Fully adjusted −log10(P value) |
|---|---|---|---|---|
| **No. of pre-existing conditions** | | | | |
| No comorbidity | 1.00 (ref) | | 1.00 (ref) | |
| 1 comorbidity | 1.15 (1.10–1.20) | 9.3 | 1.07 (1.02–1.12) | 2.1 |
| 2 comorbidities | 1.19 (1.13–1.25) | 10.2 | 1.04 (0.98–1.10) | 0.7 |
| 3 comorbidities | 1.24 (1.17–1.31) | 13.6 | 1.03 (0.97–1.10) | 0.5 |
| 4 comorbidities | 1.34 (1.26–1.42) | 21.7 | 1.08 (1.01–1.16) | 1.6 |
| ≥5 comorbidities | 1.43 (1.38–1.48) | 78.6 | 1.05 (0.97–1.13) | 0.6 |
| **Pre-existing conditions** | | | | |
| Alcohol Abuse | 1.18 (1.09–1.28) | 4 | 0.96 (0.88–1.05) | 0.4 |
| Anemia | 1.21 (1.16–1.26) | 18.6 | 1.00 (0.95–1.05) | 0 |
| Arrhythmia | 1.28 (1.23–1.33) | 33.7 | 0.98 (0.93–1.02) | 0.5 |
| Asthma | 1.16 (1.11–1.21) | 10.1 | 1.05 (0.98–1.12) | 0.7 |
| Cancer | 1.25 (1.19–1.31) | 20.7 | 1.13 (1.07–1.18) | 6 |
| Chronic kidney disease | 1.31 (1.26–1.36) | 39.6 | 1.05 (1.00–1.10) | 1.1 |
| Chronic pulmonary disorders | 1.23 (1.18–1.27) | 26.7 | 1.03 (0.96–1.10) | 0.4 |
| Cirrhosis | 1.43 (1.29–1.58) | 11.1 | 1.22 (1.10–1.35) | 3.6 |
| Coagulopathy | 1.34 (1.27–1.41) | 26.5 | 1.07 (1.01–1.14) | 1.9 |
| Congestive heart failure | 1.32 (1.26–1.38) | 32.6 | 1.04 (0.98–1.09) | 0.8 |
| COPD | 1.29 (1.22–1.38) | 15 | 0.98 (0.90–1.05) | 0.3 |
| Coronary artery disease | 1.24 (1.19–1.29) | 23.3 | 0.99 (0.94–1.04) | 0.2 |
| Dementia | 1.46 (1.37–1.56) | 27.3 | 1.03 (0.96–1.11) | 0.4 |
| Diabetes type 1 | 1.22 (1.07–1.38) | 2.7 | 1.11 (0.98–1.25) | 0.9 |
| Diabetes type 2 | 1.23 (1.18–1.27) | 30.4 | 0.99 (0.93–1.06) | 0.1 |
| End stage renal disease/dialysis | 1.23 (1.15–1.31) | 8.9 | 1.03 (0.96–1.11) | 0.4 |
| Hemiplegia | 1.20 (1.06–1.36) | 2.5 | 0.96 (0.85–1.09) | 0.3 |
| Herpes Zoster | 0.99 (0.83–1.17) | 0 | 0.89 (0.75–1.05) | 0.8 |
| HIV | 1.03 (0.92–1.16) | 0.2 | 0.93 (0.83–1.04) | 0.7 |
| Hypertension | 1.28 (1.24–1.31) | 57.2 | 1.01 (0.97–1.06) | 0.3 |
| Inflammatory bowel disorder | 0.99 (0.86–1.15) | 0 | 0.99 (0.86–1.14) | 0.1 |
| Lupus or SLE | 1.04 (0.89–1.21) | 0.2 | 0.94 (0.81–1.10) | 0.4 |
| Mental health disorders | 1.24 (1.19–1.30) | 20.4 | 1.08 (1.03–1.13) | 2.5 |
| Multiple sclerosis | 1.04 (0.85–1.27) | 0.2 | 1.03 (0.85–1.25) | 0.1 |
| Obstructive sleep apnea | 1.09 (1.03–1.16) | 2.5 | 0.98 (0.92–1.04) | 0.3 |
| Other substance abuse | 1.19 (1.12–1.26) | 8.5 | 1.06 (1.00–1.14) | 1.2 |
| Parkinson's disease | 1.15 (0.96–1.37) | 0.9 | 0.90 (0.75–1.07) | 0.7 |
| Peripheral vascular disorders | 1.23 (1.16–1.30) | 13.3 | 0.99 (0.93–1.05) | 0.1 |
| Pregnant | 0.70 (0.65–0.77) | 15.5 | 0.79 (0.72–0.86) | 6.7 |
| Pulmonary circulation disorder | 1.39 (1.30–1.49) | 19.7 | 1.09 (1.01–1.18) | 1.6 |
| Rheumatoid arthritis | 1.15 (1.03–1.29) | 1.9 | 1.04 (0.93–1.17) | 0.3 |
| Seizure/epilepsy | 1.22 (1.11–1.34) | 4.6 | 1.05 (0.96–1.16) | 0.6 |
| Severe obesity (BMI ≥ 40 kg/m²) | 1.09 (1.03–1.15) | 2.5 | 1.02 (0.96–1.08) | 0.2 |
| Sickle cell | 1.07 (0.90–1.26) | 0.3 | 1.01 (0.85–1.20) | 0 |
| Weight loss | 1.38 (1.29–1.47) | 21.5 | 1.09 (1.02–1.17) | 1.8 |
| Corticosteroids drugs | 1.09 (1.05–1.14) | 4.5 | 0.98 (0.94–1.03) | 0.3 |
| Immunosuppressant drugs | 1.01 (0.95–1.07) | 0.1 | 0.91 (0.85–0.97) | 2.4 |

*COPD* chronic obstructive pulmonary disease, *SLE* systemic lupus erythematosus.

ᵃ*Ref* reference group, *95% CI* 95% confidence interval. Risks in each categorical class were compared to the reference. −log10 (p-value), the larger, the more significant. Two reference values are −log10(0.05) = 1.3, −log10(0.01) = 2. See the PASC definition in the method section. See the adjusted covariates in the method section. Category names are in bold.

ᵇPre-existing conditions were ascertained existence if two records in the past 3 years prior to index event.

**Fig. 2 | Identified risk factors associated with incident PASC conditions from the INSIGHT cohort, March 2020 to November 2021.** The adjusted hazard ratios of different factors were reported by applying screen criteria C1–C3. The associations whose marginal increased risks than non-infected control patients were also significant regarding criteria C4 were highlighted in purple squares. See details of C1-4 in the method-association analysis section. The color bar represents different risk levels. Any PASC represents having at least one of the conditions below. The color panels represent different organ systems, including (from top to bottom): the nervous system or mental disorders, skin, respiratory system, circulatory system, endocrine and metabolic, digestive system, genitourinary system, and other signs. ICD-10 codes B948 (sequelae of other specified infectious and parasitic diseases) and U099 (post-COVID-19 condition, unspecified) were used to capture general PASC diagnoses. Source data are provided in Supplementary Data 2.

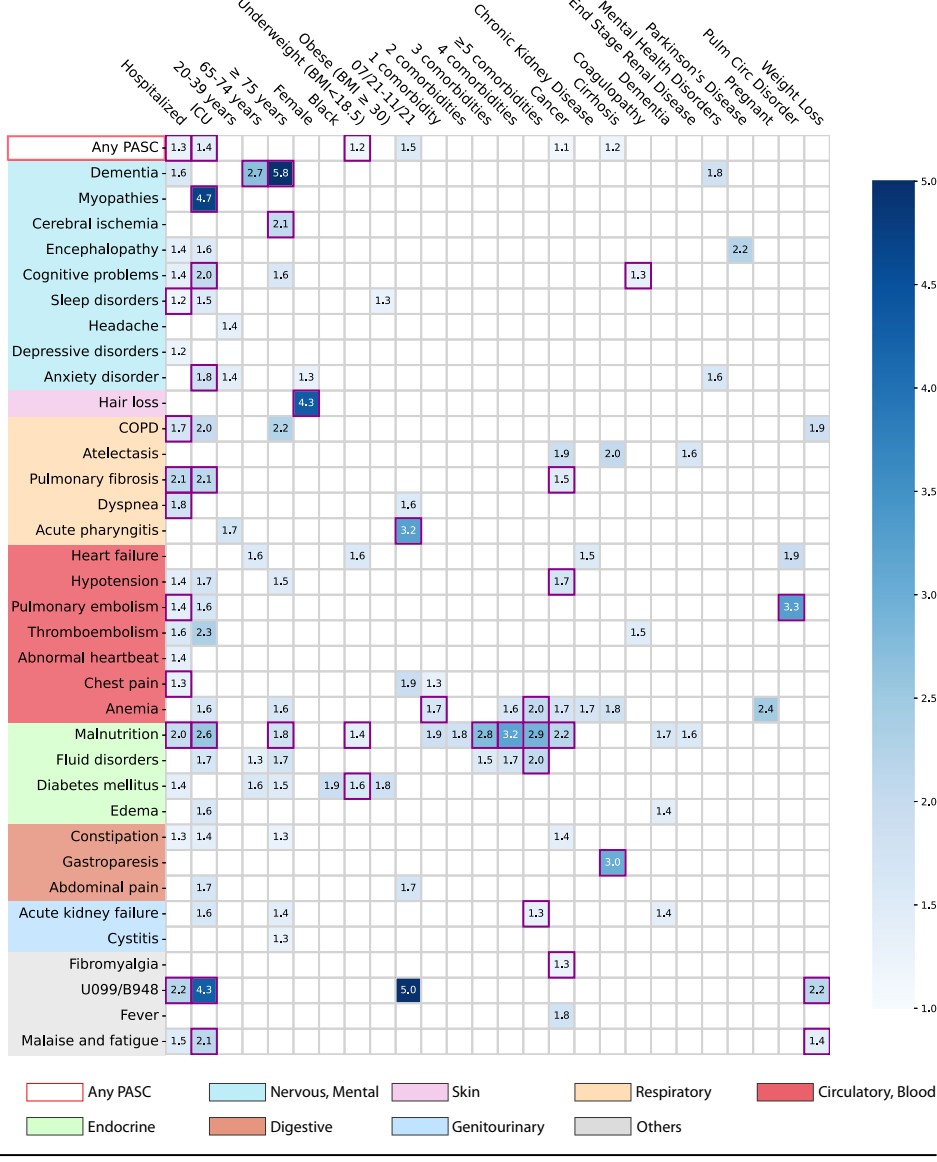

patients. Patients who were self-identified as black exhibited a 1.9-fold increased risk of being diagnosed with incident diabetes mellitus than patients who were self-identified as white, but not significant regarding Criterion 4 (method-association analysis section).

**Body mass index**

Patients who were underweight (BMI < 18.5 kg/m²) were at a 1.2-fold higher risk of being diagnosed with any incident PASC conditions than those with normal BMI (BMI from 18.5 to 24.9 kg/m²). Specifically, underweight patients were at increased risk of being diagnosed with malnutrition (1.4-fold) and diabetes mellitus (1.6-fold).

**Period of infection**

We observed that patients who got infected from July 2021 to November 2021, which was dominated by the Delta variant of SARS-CoV-2[27], showed an increased risk of being diagnosed with incident acute pharyngitis (3.2-fold) in the post-acute infection period compared to patients who got infected during March 2020–June 2020 (the 1st wave) as the reference period.

**Pre-existing conditions**

As shown in Fig. 2, having five or more baseline conditions was associated with an increased risk of potential PASC diagnoses in the post-acute phase,

including anemia (2.0-fold), fluid disorders (2.0-fold), acute kidney failure (1.3-fold) than patients without documented baseline conditions. Specifically, cancer patients showed increased risk in several post-acute conditions including atelectasis, fever, anemia, pulmonary fibrosis (1.5-fold), hypotension (1.7-fold), malnutrition (2.2-fold), and fibromyalgia (1.3-fold) compared to those without cancer diagnoses at baseline. Those with baseline pulmonary circulation disorder showed a 3.3-fold increased risk of pulmonary embolism than patients without this condition at baseline. Patients with weight loss at baseline were at a higher risk of being diagnosed with unspecified PASC diagnoses U099/B948 (2.2-fold) and malaise and fatigue (1.4-fold) than patients without the weight loss diagnosis at baseline.

**Associations stratified by acute care settings**

We further conducted analyses to examine the associations between baseline factors and incident PASC conditions among subpopulations stratified by their care settings in the acute phase (hospitalized versus non-hospitalized). The same analytics and screen criteria were used in the subgroup analyses as we did in the primary analyses. Different association patterns were observed across the two different settings as shown in Fig. 3. For patients who were not hospitalized during acute infection, being older age (≥75 years old) and having baseline cancer were associated with an increased risk of being diagnosed with a range of conditions in the post-acute period. However, patients who were hospitalized during their acute

**Fig. 3 | Identified risk factors associated with incident PASC conditions from the INSIGHT cohort, stratified by the hospitalization status during the acute infection, March 2020 to November 2021.** a non-hospitalized and (b) hospitalized during acute infection. The adjusted hazard ratios of different factors were reported by applying screen criteria C1-C3. The associations whose marginal increased risks than non-infected control patients were also significant regarding criteria C4 were highlighted in purple squares. See details of C1-4 in the Method-Association analysis section. The color bar represents different risk levels. Any PASC represents having at least one of the conditions below. The color panels represent different organ systems, including (from top to bottom): the nervous system or mental disorders, skin, respiratory system, circulatory system, endocrine and metabolic, digestive system, genitourinary system, and other signs. ICD-10 codes B948 (sequelae of other specified infectious and parasitic diseases) and U099 (post-COVID-19 condition, unspecified) were used to capture general PASC diagnoses. Source data are provided in Supplementary Data 2.

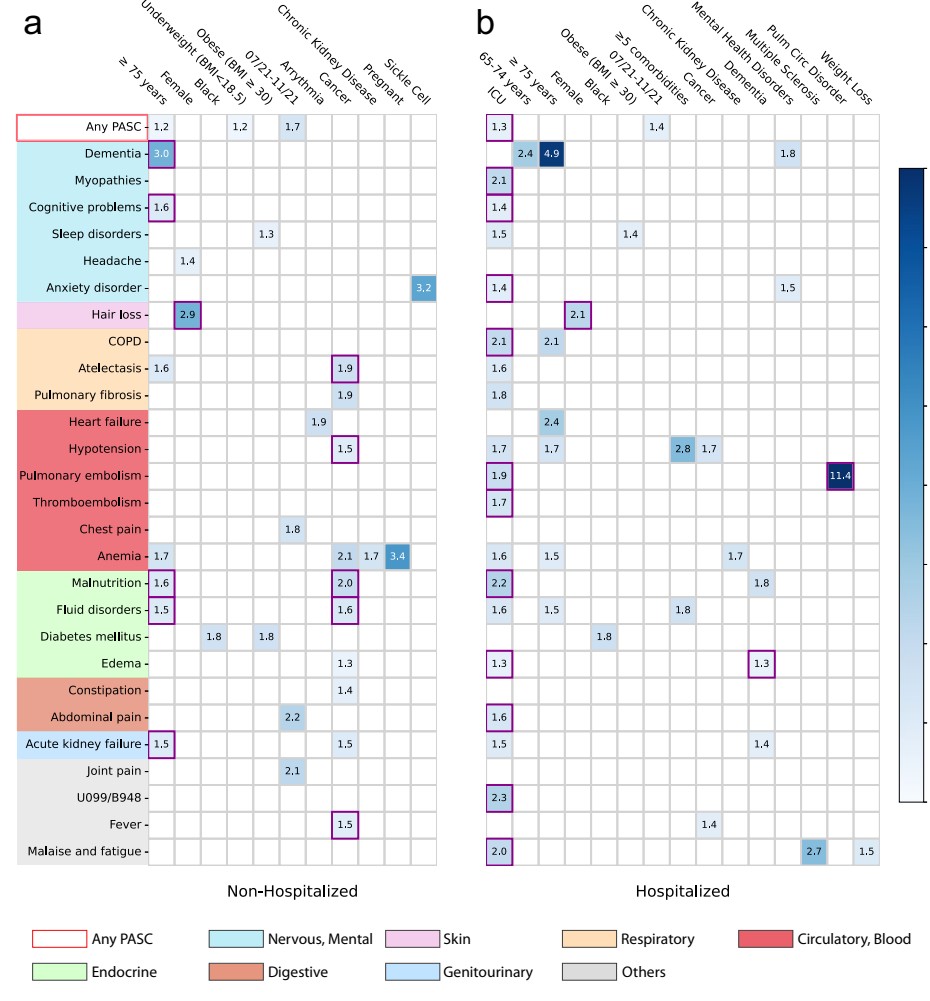

**Fig. 4 | Prediction performance of different incident potential PASC conditions from baseline characteristics and the severity in the acute phase, the INSIGHT cohort from March 2020 to November 2021.** The C-index with 95% confidence intervals as error bars was reported. Any PASC represents having at least one of the conditions below. The bars in different colors were organized by their organ systems including (from left to right): the nervous system or mental disorders, skin, respiratory system, circulatory system, endocrine and metabolic, digestive system, genitourinary system, and other signs. The conditions with a C-index in [0.8, 1) were highlighted with "o" texture, and those with a C-index in [0.7, 0.8) were highlighted with "\" texture. The numbers at the top of the bars denote the rank of the predictability quantified by the C-index among all the bars. The 95% confidence interval was estimated by 1000-times bootstrapping performance on the testing dataset in repeated cross-validation. Source data are provided in the Supplementary Data 2.

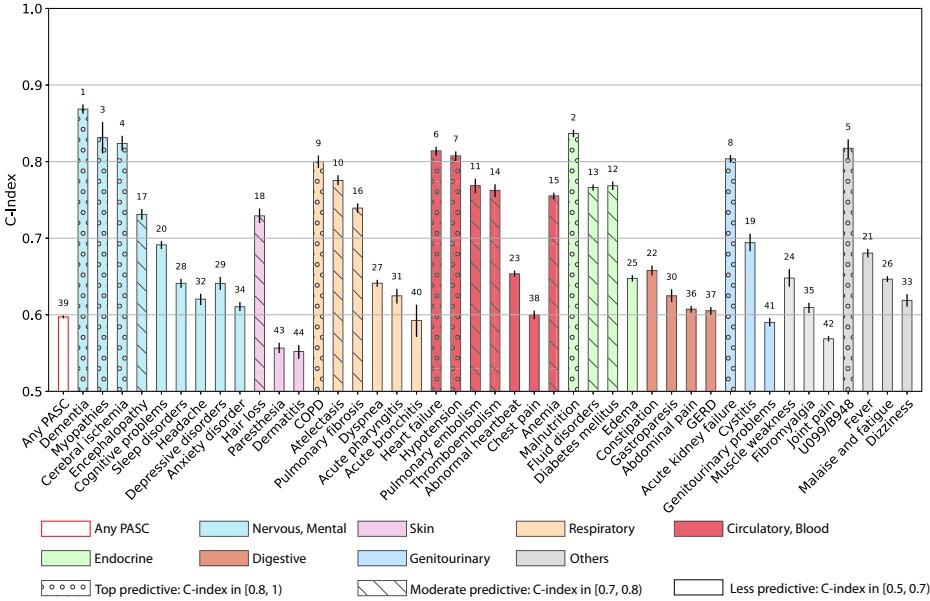

infection were in ICU or had baseline conditions including dementia or pulmonary circulation disorder were associated with an increased risk of being diagnosed with different PASC conditions. The female patients were associated with an increased risk of hair loss regardless of their acute settings.

## Prediction performance

Figure 4 summarizes the predictability of different PASC conditions, quantified by the Concordance index (C-index)[28] with a 95% confidence interval using a regularized Cox model (Method section). Similar heterogeneous predictive patterns from other machine learning models were also

observed and summarized in Supplementary Fig. 4. We observed heterogeneous predictive performance in predicting different PASC conditions: a) conditions with top predictive performance, defined as C-index ≥ 0.8, are dementia, myopathies, cerebral ischemia, COPD, heart failure, hypotension, malnutrition, acute kidney failure, and non-specific PASC diagnoses U099/B948; b) conditions with moderate predictive performance, defined as C-index in [0.7, 0.8], are encephalopathy, hair loss, atelectasis, pulmonary fibrosis, pulmonary embolism, thromboembolism, anemia, fluid disorders, diabetes mellitus; and c) other conditions such as sleep disorders, headache, depressive disorders, anxiety disorder, parethesia, dermatitis, joint pain, malaise and fatigue, and dizziness, etc, were less predictable with a C-index <0.7 or less.

### Sensitivity analysis

We have examined the impact of criteria 3 and 4 (method-association analysis section) which require the identified association to be with a higher risk in SARS-CoV-2 infected patients compared to non-infected patients. As shown in Supplementary Fig. 1, we observed that more associations have been identified if the aHR of the interaction term is smaller than 1 (without Criterion 3, highlighted in red squares) and many of these associations may not be relevant to SARS-CoV-2 infection. Taking patients with pre-existing cancer as an example, they were associated with a higher risk of being diagnosed with encephalopathy, thromboembolism, fluid disorders, edema, acute kidney failure, malaise, and fatigue in the post-acute period after SARS-CoV-2 infection. However, these associations might be identified for non-infected cancer patients as well. Therefore, criterion 3 is necessary for filtering out the associations that are not specific to SARS-CoV-2 infection. On the other hand, the identified associations including atelectasis, anemia, constipation, and fever, can be further filtered out if we require criterion 4 (highlighted in green squares in the Supplementary Fig. 1), namely the marginally increased risk of those associations to be significant (p Value < 0.05, the Wald Chi-Square test of the interaction terms in the second Cox proportional hazard model when using control patients).

We also tested to what extent the predictability of incident potential PASC conditions is affected by different machine learning models. We investigated a range of machine learning models with different complexities, including regularized logistic regression models, gradient boosting machines, and feed-forward deep neural networks. As shown in Supplementary Fig. 2, we observed similar performance of these different models, and the heterogeneous predictability patterns were still observed as in Fig. 4.

Lastly, we studied if different feature engineering can impact the prediction results of different PASC conditions. Instead of using pre-defined baseline comorbidities, we used a more high-dimensional approach by using the first three digits of ICD-10 codes of all the recorded diagnoses and prescriptions in RxNorm codes at their active ingredient level in the baseline period to predict PASC. We finally used 1593 ICD-10 diagnosis codes and 2309 drugs from the INSIGHT and 1698 ICD-10 diagnosis codes, and 4366 drugs from the OneFlorida+ data. We reported the predictive performance of different machine learning models using this large set of features in Supplementary Fig. 3, which does not show big differences compared to the performance in Supplementary Fig. 2 or main Fig. 4, and the heterogeneous predictability patterns remain the same.

### Validation by the OneFlorida+ cohort

To assess the generalizability of our findings, we replicated our analyses on the OneFlorida+ cohort as an independent validation. The OneFlorida+ cohort included 22,341 adult patients with lab-confirmed SARS-CoV-2 infection and 177,010 non-infected as control patients (See inclusion cascade in Fig. 1). We summarized the prediction performance of different potential PASC conditions with regularized Cox model in Supplementary Fig. 4 and the identified risk associations in Supplementary Fig. 5. From Supplementary Fig. 4 we again observed the heterogeneous predictability of different conditions as has been observed in Fig. 4, and the more predictable conditions (with c-index > 0.8, such as malnutrition, COPD, dementia, and acute kidney failure) and less predictable (with c-index around or below 0.6,

such as fatigue, anxiety, sleep disorders, and depression) remained the same. Similarly, the risk associations shown in Supplementary Fig. 5 are consistent with the risk associations shown in Fig. 2. Hospitalization and ICU admission in the acute infection phase were associated with a diverse set of incident diagnoses in the post-acute infection phase. We still observed the risk associations between older age and dementia (5.4-fold increased risk), female and hair loss (2.2-fold increased risk), black race, and diabetes (1.5-fold increased risk). Infection confirmation from July to November 2021 was associated with a 1.7-fold increased risk of being diagnosed with general PASC symptoms and signs (the U099/B948 ICD code).

## Discussion

In this paper, we investigated associated risk factors for a wide range of PASC conditions as well as the predictability of PASC using the EHR data from two large-scale PCORnet CRN, INSIGHT, and OneFlorida+. Compared with existing research on this topic which was mostly based on patient-reported symptoms[14,29], our study was based on routinely collected EHR datasets, aimed to uncover potentially complex association maps between baseline covariates and a set of heterogenous PASC conditions, and checked their generalizability across two different populations.

We examined the associations between a broad list of baseline covariates and a list of likely PASC conditions. The baseline covariates include demographics (age, gender, race, ethnicity, etc.), socioeconomic status, healthcare utilization, BMI, time of infection, a list of comorbidities, and severities in the acute phase of SARS-CoV-2 infection according to care settings. What distinguishes our analytic method from existing association analysis are two folds: we conducted fully adjusted association analyses for all baseline covariates and each PASC condition to uncover potentially complex association maps, and we adopted a set of stringent screening criteria to identify likely risk factors including comparing with the non-infected comparison group to remove background associations and using corrected P-value to reduce the chance of false findings in the multiple test settings. Specifically, following prior research on PASC[3,6,7,30], we ascertained newly incident conditions in the post-acute infection period (30 days to 180 days after infection) in this study. We have built a comprehensive list of diagnoses based on a prior study by Al-Aly et al.[3] with further refinements from our clinician team[6,7]. Different from existing relevant studies that treated PASC as a holistic concept[3,30], we have explored the potential risk factors of each condition, as there had been abundant evidence suggesting PASC was a highly heterogeneous condition affecting multiple organ systems[3,6,7]. Second, for a covariate to be considered as a potential risk factor of a specific condition, we required its corresponding fully adjusted hazard ratio (aHR) to be larger than 1, statistically significant in the multiple testing setting, and we further required the estimated aHR value to be larger in patients who were infected by SARS-CoV-2 compared to the non-infected patients, in this way associations that may not be attributed to COVID-19 can be filtered out (See Supplementary Fig. 1.). Figure 2 and Supplementary Fig. 5 summarized the identified risk associations from the INSIGHT and OneFlorida+ cohorts respectively. Both figures show that severe acute infection approximated by hospitalizations and admissions to the ICU during the acute infection phase was associated with a broad set of incident conditions in the post-acute infection phase, covering multiple organ systems. Older age (≥75 years) was also found to be a potential risk factor for many of these conditions. These discoveries were consistent with the conclusions from prior studies[31,32]. Other notable risk associations consistently identified from both cohorts include higher baseline comorbidity burden and fluid disorder, baseline obesity and sleep disorder, as well as baseline end-stage renal disease and malnutrition. Some associations should be interpreted more cautiously. For example, baseline pulmonary circulation disorder was consistently identified as a risk pulmonary embolism, but the two conditions are highly correlated with each other, and this association could just be due to the ICD coding and grouping. Another example was baseline pregnancy and anemia, as anemia is the most common hematologic problem in pregnancy[33]. However, there were also studies suggesting that SARS-CoV-2 infection during pregnancy can further exacerbate iron

deficiency anemia due to hyperinflammation during the acute infection phase[34]. These findings were in line with our approach which can identify associations with potentially exacerbated risks than the non-infected control group.

We then investigated the predictability of different potential PASC diagnoses using different types of machine learning (ML) models including linear models (regularized logistic regression, regularized Cox regression), gradient-boosting-tree-based models, and deep learning models, based on a similar set of baseline covariates (patient demographics, prior conditions, and care settings in the acute phase, etc.). The results from regularized Cox regression were summarized in Fig. 4, which suggested that different conditions were associated with different predictabilities in the INSIGHT cohort. Conditions such as dementia, heart failure, and kidney failure were more predictable. These conditions are with clear diagnostic criteria according to the underlying disease etiologies and are more likely to be severe COVID complications. General PASC symptoms and signs with the U099/B948 codes were also associated with good prediction performance, which is consistent with prior studies[35]. One potential reason was that these codes were relatively new, and the clinicians might be cautiously using them only when the symptoms and signs were typical. Conditions such as headache, dizziness, chest pain, joint pain, anxiety, and depressive disorders, were more difficult to predict. These conditions are most subjective to diagnose, more similar to patient-reported symptoms, and cannot be explained by alternative disease etiologies. The prediction performance obtained from more complex machine learning models or different feature engineering methods did not make such differences, as evidenced by Supplementary Figs. 2 and 3, respectively. In addition, we have replicated the predictive modeling analysis on the OneFlorida+ cohort, and the results summarized in Supplementary Fig. 4 were highly consistent with the conclusions obtained from the INSIGHT cohort. Our ML-based predictive models together with observed predictability shed light on how to use EHR data to data-drivenly identify patients who were at risk of heterogeneous PASC conditions.

There were several strengths of our study. First, we studied a comprehensive set of associations between 89 factors and 43 incident PASC conditions in two large EHR cohorts. To our knowledge, this is one of the largest studies on predictive modeling and risk factor analysis for PASC using EHR. Particularly, to reduce false findings, we adopted a non-infected control group and required the adjusted hazard ratio value of the identified association estimated from the infected patients to be larger than the value estimated from the control patients. We also identified likely associations by using significance levels corrected by multiple test settings. On the other hand, extensive sensitivity analyses and validation analyses were conducted to get robust conclusions. We derived our primary results from INSIGHT and did a validation study on OneFlorida+, which validated the generalizability of our findings. We also checked identified associations when stratifying patients by their acute severity. Regarding the prediction performance, we investigated a range of different machine learning models on both a narrow and broad list of covariates, which further validates the robustness of our conclusions.

Our study had several limitations. Our analysis was based on EHR data, which would miss the information from patients who did not visit the hospitals within the CRNs. We only considered newly incident conditions in the post-acute period but did not explore conditions that were prolonged, worsened, or relapsed before and after COVID-19 infection, as well as condition clusters or subphenotypes. The identification of the incident events can be associated with healthcare utilization behaviors: patients who had limited healthcare engagement before COVID-19 infection subsequently might have a greater opportunity to be diagnosed with new conditions simply because of less captured baseline status. Thus, in our analysis, we compared identified associations with those in non-infected patients with similar baseline characteristics including healthcare utilization behavior. In addition, we will also explore clinical notes to better capture incidence events in our future analysis. We acknowledge the limitation in not

using COVID-19 vaccination status because the publicly available COVID-19 vaccine began in early December 2020 and nearly half of the study population got infected before any vaccine was available. Regarding the remaining half of the population who got infected after December 2020, the vaccine records collected outside the hospitals were largely missing. In addition, vaccinated patients can still develop severe infection[36], which was identified as a risk factor for Long COVID by our analysis and others[37]. In addition, the effect of COVID-19 vaccine on Long COVID is not consistent and still needs further investigation[38-40]. Studying how COVID-19 vaccine influences the PASC is a promising future direction as in the later cohort vaccination is more prevalent, and building the linkage to more robust vaccination data (e.g., registry database) of general patients is one of our ongoing efforts. The smoking status was not investigated due to missingness (90.2%). In addition, our analyses did not cover the recent Omicron wave due to the availability of the data. We captured the acute severity of illness by hospitalization and ICU status during their acute infection phase, consistent with the existing Long COVID literature. However, these modelings of acute severity can lack granularity in the medical use variable that may overlook differences in the true severity of illness. For example, a patient who spent a month in the ICU on a ventilator should not be considered as having the same severity of illness as a patient who spent a night in the ICU after elective surgery. We would like to add more granularity to acute severity modeling by capturing the duration of ICU/hospital stay or using the WHO ordinal clinical severity scale[41] in our future analysis as more patients accumulate. Lastly, though we tried to remove background associations, several identified associations should be interpreted with caution. For example, older individuals are more likely to develop dementia, and those with a BMI < 18.5 are more likely to be diagnosed with malnutrition likely represents underlying patient characteristics and known disease state processes. To quantify the potential exacerbation effect (if any) of SARS-CoV-2 infection on some known risk associations remains an open question. Further studies are also warranted to investigate the basic mechanisms of developing Long COVID.

In conclusion, we used two large-scale CRN, INSIGHT and OneFlorida+ to identify likely risk factors associated with incident PASC conditions. We observed complex association patterns and a varying predictability of several PASC conditions which may represent challenges for managing heterogeneous PASC conditions. Among complex association patterns observed, we further highlighted severe acute infections, being underweight, and having baseline conditions including cancer or cirrhosis that are potentially associated with overall incident PASC in the post-acute phase. However, multiple less predictable PASC diagnoses represent an ongoing challenge that may not respond to other measures that decrease the severity of acute COVID-19. Our developed machine learning-based predictive models can help identify those who are at risk of diverse PASC conditions with heterogeneous predictability.

## Data availability

The INSIGHT data can be requested through https://insightcrn.org/. The OneFlorida+ data can be requested through https://onefloridaconsortium.org. Both the INSIGHT and the OneFlorida+ data are HIPAA-limited. Therefore, data use agreements must be established with the INSIGHT and OneFlorida+ networks. The relevant source data for each figure are provided in the Supplementary Data 2-Source Data file.

## Code availability

For reproducibility, our codes are available at https://github.com/calvin-zcx/pasc_phenotype/tree/master/prediction[42]. We used Python 3.9, python package lifelines-0.2666 for survival analysis, and scikit-learn package 1.0.2 and LightGBM package 3.3.2 for machine learning modeling.

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

## Acknowledgements

This research was funded by the National Institutes of Health (NIH) Agreement OTA OT2HL161847 (contract number EHR-01-21) as part of the Researching COVID to Enhance Recovery (RECOVER) research program. The PCORnet® Study reported in this work was conducted using PCORnet®, the National Patient-Centered Clinical Research Network. PCORnet® has

been developed with funding from the Patient-Centered Outcomes Research Institute® (PCORI®). This work was conducted through the use of data from the INSIGHT Clinical Research Network and supported in part by the Patient-Centered Outcomes Research Institute (PCORI) PCORnet grant to the INSIGHT Clinical Research Network (Grant # RI-CORNELL-01-MC). The statements presented in this work are solely the responsibility of the author(s) and do not necessarily represent the views of other organizations participating in, collaborating with, or funding PCORnet® or of the Patient-Centered Outcomes Research Institute® (PCORI®).

## Author contributions

C.Z. and F.W. proposed the initial idea. C.Z. and Y.H. designed and implemented the framework and analyzed the results. D.M. and M.G.W. set up the data infrastructure and analytics environment for INSIGHT. J.B. and E.A.S. set up the data infrastructure and analytics environment for OneFlorida+. C.Z. and J.X. preprocessed the INSIGHT and OneFlorida+ data and helped with the analysis. K.L. helped in building computational phenotyping libraries. Z.X., Y.K.Z., and Y.Y.Z. helped with the statistical analysis. D.K., A.S.N., E.J.S., R.L.R., J.P.B., J.V., M.G.W., and R.K. provided clinical inputs on data, study design, and results interpretation. C.Z. drafted the initial manuscript. F.W., E.J.S., T.W.C., and R.K. made critical revisions. All authors have provided feedback to and proofread the final version of the paper.

## Competing interests

The authors declare no competing interests.

## Additional information

[1]Department of Population Health Sciences, Weill Cornell Medicine, New York, NY, USA. [2]Division of Pulmonary and Critical Care Medicine, Weill Cornell Department of Medicine, New York, NY, USA. [3]Department of Health Outcomes Biomedical Informatics, University of Florida, Gainesville, FL, USA. [4]Department of Neurology, Weill Cornell Medicine, New York, NY, USA. [5]Center for Health Services Research, Vanderbilt University Medical Center, Nashville, TN, USA. [6]Department of Population Medicine, Harvard Pilgrim Health Care Institute, Harvard Medical School, Boston, MA, USA. [7]Louisiana Public Health Institute, New Orleans, LA, USA. ✉e-mail: few2001@med.cornell.edu

