## [Peer Review File · Communications Medicine]

Reviewers' comments:

Reviewer #1 (Remarks to the Author):

“Risk factors and predictive modeling for post-acute sequelae of SARS-CoV-2 Infection: Findings from EHR Cohorts of the RECOVER Initiative” is a retrospective cohort study involving two large datasets that aims to first identify risk factors for incident PASC conditions and then to develop models to predict who will develop these conditions. Although the authors use several advanced analytic models, the method of cohort creation is fundamentally flawed and there is limited adjustment for case-ascertainment bias, which likely explains the vast majority of findings, which are more likely to be artifact due to new and increased engagement in care than true findings. Several of the associations are quite clearly spurious (e.g., COVID as the cause of anemia during pregnancy) and suggest additional attention to potential for cause and effect is needed before moving forward. Additional clinical engagement about natural history of disease and causal pathways is strongly encouraged.

As noted above, the study is heavily influenced by case ascertainment bias and regular versus non-regular users of care. Although the number of visits prior to COVID diagnosis is measured, it is not accounted for. The importance of regular engagement of the healthcare system as a driver of findings is evident from table 1. Compared to those with COVID infections, those with PASC had fewer inpatient visits, outpatient visits, and emergency room visits during the three-year period prior to COVID diagnosis. They also had lower rates of baseline morbidities AND higher median area deprivation indices, which additionally suggests that most of the incident codes are due to new engagement with the healthcare system and not attributable to COVID-19. Methodologically, this could be addressed by matching on prior engagement with the healthcare system based on number of prior visits, rather than requiring ICD one code (without even a visit required) to measure “incident cases of PASC.” (This same methodologic error is present in many of the PASC studies, including those that are heavily referenced in the paper).

In addition, the authors identify risk factors for severe COVID-19 (e.g., COPD, dementia, ICU, older age) and it is not clear the extent to which these identified associated are due to PASC or due to risk of severe disease \diamond healthcare engagement/ICU admission \diamond new case identification. Complete re-analysis with improved adjustment for healthcare engagement and timing of new diagnosis would be necessary to address this major limitation. Two articles that describe risk factors for severe disease despite vaccination are:

[https://www.thelancet.com/journals/lancet/article/PIIS0140-6736\(22\)01656-7/fulltext](https://www.thelancet.com/journals/lancet/article/PIIS0140-6736(22)01656-7/fulltext)

<https://jamanetwork.com/journals/jamanetworkopen/fullarticle/2797495>

Some specific examples of likely spurious associations for attribution to COVID/PASC (there are many, this is not an exhaustive list):

- Diagnosis of anemia following COVID diagnosis in pregnant patients (this is likely due to new engagement in care, and the expected outcome of pregnancy, which is birth)
- Association between low BMI, COVID, and malnutrition. The malnutrition diagnosis is almost certainly secondary to the low BMI, not the incident COVID diagnosis.
- Association between cancer and malnutrition (cancer and chemotherapy cause malnutrition)
- Association between Parkinson's disease and encephalopathy
- CKD and heart failure/anemia (CKD causes anemia)

- Association between obesity and diabetes (again, obesity causes diabetes)

Given the fundamental methodologic concerns noted above, I will not make specific comments about the tables, figures, etc, as these would require complete revision after re-analysis in order for this article to be suitable for publication.

However, as structured, several sections of the manuscript are quite repetitive (E.g., methods and results section contain close to the same paragraph) and the manuscript as a whole could be reduced by ~25% without losing any content. Additionally, the major focus of the results section is on the associations listed above, rather than the predictive models. After reading the results section, it is not clear what is included in the models and whether those models would be clinically useful tools for identifying risk. Strongly suggest removing the redundancy and including more detail about model findings and how they could be used to inform clinical practice.

Reviewer #2 (Remarks to the Author):

In their manuscript, "Risk Factors and Predictive Modeling for Post-Acute Sequelae of SARS-CoV-2 Infection: Findings from EHR Cohorts of the RECOVER Initiative," Zang et al describe associations between EHR derived conditions and the risk of various PACS conditions/symptoms. The authors should be applauded for tackling such a complex issue and attempting to do so using EHR data, which tends to lend itself to broader use once validated. There are several limitations which warrant further evaluation. In particular, despite excellent efforts to control for it, residual confounding likely remains, driven in part by changes in engagement with the healthcare system following SARS-CoV-2 infection. Further, the outcomes require additional clinical context for, as the authors note, several of the associations are among conditions that are themselves highly correlated with each other.

1. Number of patients listed in abstract is different than the numbers listed in lines 117-118. Further, saying that there were 12 and 15 million patients in each cohort is misleading as there were far less in the final cohorts, as indicated in Figure 1.
2. Given the degree of missingness for smoking (90.2%) noted on line 184, is it worth even including this variable in the model?
3. How was death within 30 days of diagnosis handled?
4. Several other important variables should be considered including the repeat/multiple infections, vaccination status (mentioned in the discussion, but a major limitation given data indicating that this may reduce severity of illness and PASC risk), and treatment with SARS-CoV-2 specific therapies.
5. Is the discharge location of patients known? This is of particular importance in the development of conditions such as pressure ulcers, which are more likely to occur in individuals living in care facilities.
6. Severity of illness is clearly a driving factor for PACS risk, particularly myopathies, pressure ulcers, malnutrition, and U099/B948 codes. With the exception of the latter, however, ICU stays (particularly long stays) place individuals at increased risk for these conditions regardless of Covid status. While the authors worked to control for this with their non-infected group, the lack of granularity in the medical use variable may overlook differences in true severity of illness. For example, a Covid patient who spent a month in the ICU on a ventilator would be considered as having the same severity of illness as a non-Covid patient who spend a night in the ICU after an elective surgery. Was consideration given to using a more granular variable, such as the WHO ordinal

clinical severity scale?

7. Lines 289-306 are redundant from the methods and should be considered for removal or merger with the methods section.

8. In Table 1, calculation of percentages of the total for number of patients with Covid and percentages of the stratum among those with PASC makes interpretation of the true prevalence of these conditions in those with PACS difficult. Consider calculating both as percent of total for each column.

9. In Extended Table 1, the HR for PACS declines with increasing number of outpatient visits prior to infection. Did the authors consider that this may be a reflection that patients with greater engagement with the healthcare system prior to infection received diagnoses before infection (either resulting in or because of greater engagement with healthcare providers); conversely, patients who had limited healthcare engagement prior to Covid infection subsequently had greater opportunity to be diagnosed with new conditions, simply because of greater engagement?

10. Please clarify the methods for the sensitivity analysis using a 'data-drive approach.' The authors state that they used the first 3 digits of ICD-10 codes and medications codes. It is unclear what was actually done and how this is of benefit over a theory driven approach.

Reviewer #3 (Remarks to the Author):

Authors used data from two large PCORNet data sets to explore which risk factors may be associated with the occurrence of post-acute sequelae SARS-CoV-2 infection (PASC).

Major comments:

- Since this study is observational in nature, causal inferences should not be implied. Several phrases need to be edited to that end. For example, line 105 "will likely improve" implies causal inference. Similar edits need to happen in the last paragraph of Discussion.
- To define PASC, authors use the incidence of a diagnosis from a long list of conditions (from depression and anxiety to myopathies and acute kidney failure) from 31 to 180 days after COVID infection. This will likely include a lot of false positives (people can have those conditions, regardless of COVID too). They capture all of these using ICD-10 codes. The only ICD-10 codes that are specific (though not necessarily sensitive) for PASC are U099 and perhaps B948. Authors should do a sensitivity analysis, in which they only use these two diagnoses to capture the outcome of PASC. This is important because some of the risk factors they are claiming to be related to PASC (such as obesity) are indeed risk factors for COVID-unrelated incidence of those conditions in the long list, so using a narrow, more specific definition for PASC may validate (or refute) if these risk factors are indeed related to PASC.
- Authors do try to address the above concern using a second Cox model that include all patients. However, in lines 216-221 they only require the direct association to be statistically significant, and they don't seem to enforce statistical significance for the confirmatory interaction term in the second model. The results of this model with interaction terms are also not fully reported; I strongly suggest including that as a table in the supplement. In fact, one could argue that the risks reported in lines 337-397 should be replaced with those calculated on the interaction terms in the second model. Right now, it seems like these risks are from the first model, so the fact that obese patients had a 1.4-fold increased risk of diabetes may not describe anything about COVID-related diabetes and be merely a representation of the (well known) association between obesity and diabetes.
- On line 200 authors state that "loss of follow-up" is used as a censoring event. However, their cohort, by definition, only includes patients who were followed up (because cohort definition

requires at least one diagnosis code in days 31 to 180 after COVID incidence, which is defined as “follow up period” in line 198). Authors need to explain how they are defining loss of follow-up.

- Lines 298-309 really belong in Methods, not results. Lines 322-328 are repeating what was already stated in Methods.
- More inpatient or outpatient visits in the past 3 years seem to be negatively associated with PASC. How do authors explain this?
- Later infection periods (after 3/21) seem to be positively associated with PASC. How do authors explain this?
- Prior research has repeatedly suggested that obesity is associated with PASC. In this study, it doesn't appear associated, and in fact, being underweight is associated with PASC. How do authors explain this?

Minor comments:

- The title should be revised. “Predictive modeling” is an action phrase, so the first phrase should match. Perhaps “Identifying risk factors and predictive modeling of post-acute ...”?
- Line 76, for citations 3 and 5 both, page 19 is referenced. Citation 3 has page numbers that don't include 19 (they range from 259 to 264), and citation 5 is a 13-page document. Please correct citations.
- Line 89, the phrase “systematic study” seems like puffery. “Study” should suffice.
- Line 93, “used as” should be changed to “used for”.
- Citation 14 should include “medRxiv” in it.
- Line 115, “real-world” seems like puffery. EHR data is typically from real world. There is no need to emphasize that.
- The first table in the Supplement (pages 1-138 of the supplement) appears to have a lot of duplicative rows. Was it exported correctly? Should it be de-duplicated? Looking at pages 277 and beyond, I wonder if this was supposed to be seen as a table with five columns, but the way it was exported makes it difficult to align the cells that are on the same row. Can it be re-exported, perhaps in a “wide” format (landscape, as opposed to portrait)?
- Line 230, the word “in” is missing before “a predefined hyperparameter space”.
- Line 250 mentions the first “3-digits” of ICD-10 codes. ICD-10 codes are not numbers; they in fact start with a letter. Either authors mean the “first 3 characters”, or they may “the first 4 characters” (letter + 3 digits). This should be clarified. Moreover, ICD-10 hierarchy doesn't always work this way, i.e., the first 3 characters are not always going to give you a roll up of equal granularity. Authors should justify their choice.
- Line 284, “has” should be changed to “have”.
- Lines 321-322 include an incomplete sentence.
- The GitHub link on lines 554-555 does not work.

Referee expertise:

Referee #1: MD in Infectious Disease, epidemiology

Referee #2: MD in cardiology, healthcare technology, electronic health records

Referee #3: MD, biomedical informatics

Reviewers' comments:

We appreciate the thoughtful comments from the reviewers and provide our point-by-point response in this revision as below.

Reviewer #1 (Remarks to the Author):

"Risk factors and predictive modeling for post-acute sequelae of SARS-CoV-2 Infection: Findings from EHR Cohorts of the RECOVER Initiative" is a retrospective cohort study involving two large datasets that aims to first identify risk factors for incident PASC conditions and then to develop models to predict who will develop these conditions. Although the authors use several advanced analytic models, **the method of cohort creation is fundamentally flawed and there is limited adjustment for case-ascertainment bias, which likely explains the vast majority of findings, which are more likely to be artifact due to new and increased engagement in care than true findings.** Several of the associations are quite clearly spurious (e.g., COVID as the cause of anemia during pregnancy) and suggest additional attention to potential for cause and effect is needed before moving forward. **Additional clinical engagement about natural history of disease and causal pathways is strongly encouraged.**

As noted above, the study is heavily influenced by **case ascertainment bias and regular versus non-regular users of care. Although the number of visits prior to COVID diagnosis is measured, it is not accounted for.** The importance of regular engagement of the healthcare system as a driver of findings is evident from table 1. Compared to those with COVID infections, those with PASC had fewer inpatient visits, outpatient visits, and emergency room visits during the three-year period prior to COVID diagnosis. They also had lower rates of baseline morbidities AND higher median area deprivation indices, which additionally

suggests that **most of the incident codes are due to new engagement with the healthcare system and not attributable to COVID-19**. Methodologically, this could be addressed by matching on prior engagement with the healthcare system based on number of prior visits, rather than requiring ICD one code (without even a visit required) to measure “incident cases of PASC.” (This same methodologic error is present in many of the PASC studies, including those that are heavily referenced in the paper).

Response: Thanks for the comments. We would like to do the following clarifications.

Regarding the engagement concerns, first, we not only measured but also adjusted for the history of healthcare engagement covariates in our association analysis (Methods-covariates and we further clarified these in the discussion section). Second, Regarding incident PASC codes, in our previous literature, we also adopted IPTW ¹ and matching ² to account for the engagement. Third, compared with existing association analysis literature, we further compared the non-infected group and required the identified association should be enhanced compared with the non-infected group. Because the bias of the new engagement argument can also hold for non-infected groups and our approach might mitigate your proposed concerns (Methods-Association analysis-step II).

Regarding the potentially spurious associations. One example was baseline pregnancy and anemia, as anemia is the most common hematologic problem in pregnancy. However, there were also studies suggesting that SARS-CoV-2 infection during pregnancy can further exacerbate iron deficiency anemia due to hyperinflammation during the acute infection phase³ (in our discussion section). What distinguishes our work from existing association literature is our efforts to get rid of these spurious associations. First, we required additional/marginal association compared with non-infected patients (Method-association analysis section). Also taking anemia in pregnancy as an example, we observed worsened associations than the non-infected control group (Fig 1) and we removed a lot of potentially spurious associations as shown in the sensitivity analysis (Extended Fig 1). Second, we also used the corrected P-value (<0.000562) to identify likely associations, aiming to adjust false/spurious findings in the multiple test settings (Method-association analysis section). We identified our work as an association hypothesis generation work, which can facilitate further mechanism studies to confirm the risk factors.

In addition, the authors identify risk factors for severe COVID-19 (e.g., COPD, dementia, ICU, older age) and it is not clear the extent to which these identified associated are due to PASC or due to risk of severe disease \diamond healthcare engagement/ICU admission \diamond new case identification. Complete re-analysis with improved adjustment for healthcare engagement and timing of new diagnosis would be necessary to address this major limitation. Two articles that describe risk factors for severe disease despite vaccination are:

[https://www.thelancet.com/journals/lancet/article/PIIS0140-6736\(22\)01656-7/fulltext](https://www.thelancet.com/journals/lancet/article/PIIS0140-6736(22)01656-7/fulltext)
<https://jamanetwork.com/journals/jamanetworkopen/fullarticle/2797495>

Response: Thanks for the comments. We would like to do the following clarifications.

We also hypothesized that the severity during acute infection were factors driving PASC risks. First, we tried to approximate acute severity by their hospitalization, ICU, and ventilation status (1 day before to 16 days after the acute infection) and quantified their associations with a range of incidence conditions in the post-acute phase (30 days to 180 days after the infection). As shown in Fig1, the severe acute infection was associated with incident COPD and dementia. Second, we further conducted a subgroup analysis by stratifying patients by their severity in the acute infection phase (hospitalized or non-hospitalized) and then performing statistical analysis within each stratum. The noninfected control patients were also stratified according to the hospitalized or non-hospitalized during the acute period, to capture background associations within each subgroup population. We can see different association patterns across both two groups, as summarized in Fig 3, and Associations stratified by the acute care settings section. We found acute ICU status was also associated with incident post-acute COPD condition.

As we have replied in our previous response, we have adjusted for baseline healthcare engagement in our analysis, and further using non-infected control patients and corrected P-values to eliminate the concerns of spurious findings.

Some specific examples of likely spurious associations for attribution to COVID/PASC (there are many, this is not an exhaustive list):

- Diagnosis of anemia following COVID diagnosis in pregnant patients (this is likely due to new engagement in care, and the expected outcome of pregnancy, which is birth) (we required riskier than control group)
- Association between low BMI, COVID, and malnutrition. The malnutrition diagnosis is almost certainly secondary to the low BMI, not the incident COVID diagnosis.
- Association between cancer and malnutrition (cancer and chemotherapy cause malnutrition)
- Association between Parkinson's disease and encephalopathy
- CKD and heart failure/anemia (CKD causes anemia)
- Association between obesity and diabetes (again, obesity causes diabetes)

Given the fundamental methodologic concerns noted above, I will not make specific comments about the tables, figures, etc, as these would require complete revision after re-analysis in order for this article to be suitable for publication.

Response: Thanks for the comments. We have detailed our methods for reducing likely spurious associations in our first response, briefly including additional/marginal association compared with non-infected patients, fully adjustment for baseline covariates (including the mentioned healthcare utilization), and corrected P-values for multiple testing. The above-mentioned associations were

worsened due to covid infection compared with associations in non-infected patients by following our approach. For associations not exhibiting worsening associations were summarized in Extended data fig 1.

However, as structured, several sections of the manuscript are quite repetitive (E.g., methods and results section contain close to the same paragraph) and the manuscript as a whole could be reduced by ~25% without losing any content. Additionally, the major focus of the results section is on the associations listed above, rather than the predictive models. After reading the results section, it is not clear what is included in the models and whether those models would be clinically useful tools for identifying risk. Strongly suggest removing the redundancy and including more detail about model findings and how they could be used to inform clinical practice.

Response: Thanks for the comments. We have revised the method and results sections as suggested.

Reviewer #2 (Remarks to the Author):

In their manuscript, "Risk Factors and Predictive Modeling for Post-Acute Sequelae of SARS-CoV-2 Infection: Findings from EHR Cohorts of the RECOVER Initiative," Zang et al describe associations between EHR derived conditions and the risk of various PACS-conditions/symptoms. The authors should be applauded for tackling such a complex issue and attempting to do so using EHR data, which tends to lend itself to broader use once validated. There are several limitations which warrant further evaluation. In particular, despite excellent efforts to control for it, residual confounding likely remains, driven in part by changes in engagement with the healthcare system following SARS-CoV-2 infection. Further, the outcomes require additional clinical context for, as the authors note, several of the associations are among conditions that are themselves highly correlated with each other.

1. Number of patients listed in abstract is different than the numbers listed in lines 117-118. Further, saying that there were 12 and 15 million patients in each cohort is misleading as there were far less in the final cohorts, as indicated in Figure 1.

Response: Thanks for the comments. We have revised our abstract. Also, we would like to do the following clarifications. These are the base population in our two Clinical Research Networks and we described these base populations in the method section (also in the top rectangles in Fig1). As you mentioned, the final selected cohorts (the bottom rectangles in Fig1) were much smaller and we explicitly show the selection details in the results section and Fig 1.

2. Given the degree of missingness for smoking (90.2%) noted on line 184, is it worth even including this variable in the model?

Response: Thanks for the comments. We have revised our primary analyses by excluding the smoking variables and we also acknowledged this limitation due to the missingness in the discussion section.

3. How was death within 30 days of diagnosis handled?

Response: Thanks for your questions. We would like to do the following clarifications. Because of the intrinsic definition of the post-acute sequelae of SARS-CoV-2 infection (PASC), we selected patients who had at least one diagnosis code from 31 days to 180 days after the index date (referred to as the post-acute phase or follow-up period), to ensure that patients were connected to the healthcare system, were being observed during the study period, and were being alive beyond the acute phase. This is a criterion adopted in RECOVER initiative to screen the PASC signals.^{1,2,4} Thus, the patients who died in the acute phase were typically not included in the PASC/Long Covid studies.

4. Several other important variables should be considered including the repeat/multiple infections, vaccination status (mentioned in the discussion, but a major limitation given data indicating that this may reduce severity of illness and PASC risk), and treatment with SARS-CoV-2 specific therapies.

Response: Thanks for your questions. We would like to do the following clarifications. First, we didn't use vaccination status because a) the vaccine began in early December 2020 and more than half of the patients got infected before the vaccine was available. Even for patients after December 2020, the number of recorded patients was limited, please see the table below. We defined the fully vaccinated as two shots of mRNA vaccine (Pfizer, or Moderna) or one shot of J&J, see <https://www.cdc.gov/coronavirus/2019-ncov/vaccines/stay-up-to-date.html>. In all, the vaccine baseline covariates before the index date only accounted for 2% of the population in the covid+ group and 5% in the covid- group, barely changing the final screened conditions considering their significant hazard ratio. We acknowledged this as a limitation in the discussion section.

	INSIGHT COVID+		INSIGHT COVID-	
	N	%	N	%
Total number	61305	100%	577174	100%
Fully vaccinated - Pre-index	811	1%	17229	3%
Partially vaccinated - Pre-index	823	1%	11753	2%
No evidence - Pre-index	59671	97%	548197	95%

Second, Studying how the vaccine influences the risk of long Covid is a huge topic by itself and left as a future study with a more sophisticated experiment design, and also relies on the ongoing efforts of cumulating and collecting more vaccination data (e.g. registry database, which is not available yet).

5. Is the discharge location of patients known? This is of particular importance in the development of conditions such as pressure ulcers, which are more likely to occur in individuals living in care facilities.

Response: Thanks for your comments. As shown in Fig 3 (subgroup/stratified analysis by the hospitalization status), we investigated factors associated with incident PASC when we stratified patients according to their hospitalization status during their acute infection. We observed pressure ulcers were associated with ICU, >=75 ages, weight loss at baseline for hospitalized patients only, but not for the non-hospitalized patients. These findings were aligned with your proposed comments that pressure ulcers were more likely to occur in individuals living in care facilities.

6. Severity of illness is clearly a driving factor for PACS risk, particularly myopathies, pressure ulcers, malnutrition, and U099/B948 codes. With the exception of the latter, however, ICU stays (particularly long stays) place individuals at increased risk for these conditions regardless of Covid status. While the authors worked to control for this with their non-infected group, the lack of granularity in the medical use variable may overlook differences in true severity of illness. For example, a Covid patient who spent a month in the ICU on a ventilator would be considered as having the same severity of illness as a non-Covid patient who spend a night in the ICU after an elective surgery. Was consideration given to using a more granular variable, such as the WHO ordinal clinical severity scale?

Response: Thanks for your comments. We would like to do the following clarifications. First, we use hospitalization status, ICU, and ventilation status during the 1 day prior to 16 days after their index date to capture the severity of illness during the acute phase. See a copy of top rows of table1 below. These modeling of acute severity were consistent with our previous literature ^{1,2,4,5} and CDC work⁶.

Characteristics	Number of Patients with Lab-Confirmed SARS-CoV-2 Infection (columns %)	Number of Patients with Lab-Confirmed SARS-CoV-2 Infection and with at least One Incident PASC Condition (columns %)
Total	35,275	17,571 (49.8)
The Severity of Acute Infection — no. (%)		
Not hospitalized	22,148 (62.8)	9,809 (55.8)
Hospitalized w/o ICU	11,480 (32.5)	6,611 (37.6)
ICU	1,647 (4.7)	1,151 (6.6)

Second, we conducted a stratified analysis by stratifying patients by their severity in the acute infection phase (hospitalized or non-hospitalized) and then performing statistical analysis within each stratum. The noninfected control patients were also stratified according to the hospitalized or non-hospitalized during the acute period, to capture background associations within each subgroup population. We can see different association patterns across both two groups, as summarized in Fig 3, and Associations stratified by the acute care settings section. Third, we acknowledge potential granularity issues of acute severity and a small number of ICU patients (a limited number of Covid patients who spent a month in the ICU) in the discussion section and we would like to explore more fine-grained severity scales, like the WHO ordinal clinical severity scale, in our future analysis as more patients were accumulated.

7. Lines 289-306 are redundant from the methods and should be considered for removal or merger with the methods section.

Response: Thanks for your comments. We have removed redundant content and systematically reorganized the method and results sections.

8. In Table 1, calculation of percentages of the total for number of patients with Covid and percentages of the stratum among those with PASC makes interpretation of the true prevalence of these conditions in those with PACS difficult. Consider calculating both as percent of total for each column.

Response: Thanks for your suggestions. We have revised the table 1 by calculating percentage within each column.

9. In Extended Table 1, the HR for PACS declines with increasing number of outpatient visits prior to infection. Did the authors consider that this may be a reflection that patients with greater engagement with the healthcare system prior to infection received diagnoses before infection (either resulting in or because of greater engagement with healthcare providers); conversely, patients who had limited healthcare engagement prior to Covid infection subsequently had greater opportunity to be diagnosed with new conditions, simply because of greater engagement?

Response: Thanks for your comments. We would like to do the following clarifications. It's possible, and thus, we tried to screen likely associations by a) adjusting for a range of baseline covariates including these baseline hospitalization utilization features, and b) further compared these associations in both infected patients versus non-infected patients. The same arguments can be applied to the non-infected patients and by requiring excess risk over this control group can further mitigate the proposed concern.

10. Please clarify the methods for the sensitivity analysis using a 'data-drive approach.' The authors state that they used the first 3 digits of ICD-10 codes and medications codes. It is unclear what was actually done and how this is of benefit over a theory driven approach.

Response: Thanks for your comments. We would like to do the following clarifications. In our primary analyses, we used a list of clinician-selected baseline clinical features^{1,2,4} (theory-driven approach, relatively low dimensional) for association analysis and predictive modeling. In addition, we also check more high-dimensional baseline covariates by considering all the recorded ICD-10 codes (aggregated by their first 3-characters) and prescriptions (aggregated by their active ingredients). The use of different feature engineering methods is aimed to validate the robustness of the predictability of PASC.

Reviewer #3 (Remarks to the Author):

Authors used data from two large PCORNet data sets to explore which risk factors may be associated with the occurrence of post-acute sequelae SARS-CoV-2 infection (PASC).

Major comments:

- Since this study is observational in nature, causal inferences should not be implied. Several phrases need to be edited to that end. For example, line 105 "will likely improve" implies causal inference. Similar edits need to happen in the last paragraph of Discussion.

Response: Thanks for the comments. We have deleted this implication in line 105 and revised the discussion section accordingly.

- To define PASC, authors use the incidence of a diagnosis from a long list of conditions (from depression and anxiety to myopathies and acute kidney failure) from 31 to 180 days after COVID infection. This will likely include a lot of false positives (people can have those conditions, regardless of COVID too). They capture all of these using ICD-10 codes. The only ICD-10 codes that are specific (though not necessarily sensitive) for PASC are U099 and perhaps B948. Authors should do a sensitivity analysis, in which they only use these two diagnoses to capture the outcome of PASC. This is important because some of the risk factors they are claiming to be related to PASC (such as obesity) are indeed risk factors for COVID-unrelated incidence of those conditions in the long list, so using a narrow, more specific definition for PASC may validate (or refute) if these risk factors are indeed related to PASC.

Response: Thanks for your comments. We would like to do the following clarifications. First, we have used the general PASC codes U099/B948 as one outcome in our PASC list and we have reported the association results regarding U099/B948 codes in our results section. As shown in Figure 1, patients with weight loss or obesity (BMI ≥ 30), or who got hospitalized or in ICU during their acute period were at a higher risk of being diagnosed with these general PASC codes. Second, the U099 codes were effective since 10/1/2021, which can not capture patients who got infected before that. Third, defining PASC is still an open question and our PASC conditions were consistent with our previous literature ^{1,2,4,5}, which usually exhibited excess burden over the non-infected control group. Regarding association analyses, to reduce the change of false findings, we first adopted non-infected control groups and required the adjusted hazard ratio value of the identified association estimated from the infected patients to be larger than the value estimated from non-infected patients, and we further adopted very stringent significance level corrected due to the multiple test settings.

- Authors do try to address the above concern using a second Cox model that include all patients. However, in lines 216-221 they only require the direct association to be statistically significant, and they don't seem to enforce statistical significance for the confirmatory interaction term in the second model. The results of this model with interaction terms are also not fully reported; I strongly suggest including that as a table in the supplement. In fact, one could argue that the risks reported in lines 337-397 should be replaced with those calculated on the interaction terms in the second model. Right now, it seems like these risks are from the first model, so the fact that obese patients had a 1.4-fold increased risk of

diabetes may not describe anything about COVID-related diabetes and be merely a representation of the (well known) association between obesity and diabetes.

Response: Thanks for your comments. We have reported new results of aHR of the interaction term in Extended Data Fig 2 (compared with primary results in Fig 2), and conditions if we further required interaction terms to be significant in Extended Data Fig 3, and reported these new results in the sensitivity analysis section. Existing literature majorly reported their results based on model 1 (Level 1), and we required more stringent screening criteria in our analysis (aHR in model 1, $p\text{Value} < 0.000562$ under multiple test setting, and aHR in model 2 > 1 , denoted as Level 2). We reported our primary results in Fig 2 and the previous method in Extended Data Fig 1. as a sensitivity analysis. In addition, we further reported results if pValue of aHR in model 2 is significant in Extended Data Fig 3 as another sensitivity analysis (Level 3). By using more and more stringent criteria, we can expect potentially fewer identified associations in Fig 2 than in Extended Data Fig 1, and fewer associations in Extended Data Fig 3 than in Fig 2. This is a trade-off between sensitivity and precision, and because current knowledge is based on Level 1, and thus we reported Level 2 as a primary analysis and Level 3 as a sensitivity analysis. We identified our work as a data-driven association analysis/hypotheses generation work, which can be helpful for further mechanism study to confirm risk factors for the long covid.

- On line 200 authors state that “loss of follow-up” is used as a censoring event. However, their cohort, by definition, only includes patients who were followed up (because cohort definition requires at least one diagnosis code in days 31 to 180 after COVID incidence, which is defined as “follow up period” in line 198). Authors need to explain how they are defining loss of follow-up.

Response: Thanks for the comments. We would like to do the following clarifications. First, we required any diagnosis in days between 31 to 180 to require that a patient was alive beyond the acute phase, and he/she was connected to the EHR system and got the health status captured. This is a criterion adopted in RECOVER initiative to screen the PASC signals.

Second, the loss of follow-up in the database is used to define a censoring event. “The censoring event is defined as the earliest event of documented death, loss of follow-up in the database (the date of the last documented records in the EHR systems), 180 days after the baseline, or the end of our observational window (December 31, 2021).” For example, if a patient’s records ended at, say day 90, without documented PASC conditions, we would say that this patient lost follow-up at day 90 in the EHR databases, namely event time was ≥ 90 .

Third, we added a short description “(the date of the last documented records in the EHR systems)” to further explain the loss of follow-up in the database.

- Lines 298-309 really belong in Methods, not results. Lines 322-328 are repeating what was already stated in Methods.

Response: Thanks for your great comments. We have reorganized the result section and removed duplicated method contents.

- More inpatient or outpatient visits in the past 3 years seem to be negatively associated with PASC. How do authors explain this?

Response: Thanks for the comments. We would like to do the following clarifications. First, we didn't find significant associations between past healthcare utilization and a list PASC conditions as shown in Fig 2 (the association map in our primary results, which were fully adjusted, background signal removed, and significant under multiple tests). Second, the associations of past healthcare utilization and having at least one PASC condition were summarized in Table 2. In terms of univariate analysis, we found slightly increased HR as the number of past inpatient visits increase. However, when adjusted for all the covariates we found you mentioned "negative" associations. A possible reason is we accounted for other baseline covariates including a list of comorbidities for which the hospital visits were just proxy. Thus, we adopted very stringent screen criteria/methods to identify more likely/direct risk associations and reported them in Fig 2.

Covariate	Univariate HR (95% CI)	Univariate - log10 (P-Value)	Fully adjusted HR (95% CI)	Fully adjusted - log10 (P-Value)
No. of hospital visits in the past 3 yrs				
Inpatient 0	1.00 (ref)		1.00 (ref)	
Inpatient 1-2	1.16 (1.12-1.20)	15.3	0.96 (0.92-1.00)	1.3
Inpatient ≥ 3	1.17 (1.11-1.23)	8.6	0.87 (0.82-0.93)	4.1
Outpatient 0	1.00 (ref)		1.00 (ref)	
Outpatient 1-2	0.71 (0.66-0.76)	20.3	0.80 (0.74-0.86)	9
Outpatient ≥ 3	0.62 (0.58-0.66)	57.7	0.64 (0.60-0.68)	47.1

- Later infection periods (after 3/21) seem to be positively associated with PASC. How do authors explain this?

Response: Thanks for the comments. We would like to do the following clarifications. We have reported the associations of later infection periods (compared with the early period 3/20 to 6/20 as reference) in the result section Period of infection. This later period was dominated by the Delta variant of SARS-CoV-2, and as shown in our primary results Fig 2, we observed an increased risk of being diagnosed with incident pharyngitis (3.2-fold), chest pain (1.9-fold), abdominal pain (1.7-fold), dyspnea (1.6-fold), as well as being diagnosed with general PASC symptoms and signs with the U099/B948 ICD codes (5-fold) in the post-acute infection period compared to patients got infected during the 1st wave as the reference period. We think these results implied the potentially different PASC manifestations and associations for different variants of concerns over different periods.

- Prior research has repeatedly suggested that obesity is associated with PASC. In this study, it doesn't appear associated, and in fact, being underweight is associated with PASC. How do authors explain this?

Response: Thanks for the comments. We would like to do the following clarifications. We discussed the associations between BMI and a range of PASC conditions in the result section Body Mass Index. As shown in our primary result in Fig 2, we found BOTH obese and underweight were at higher risk of being diagnosed with a certain but different set of PASC conditions. Specifically, underweight patients were at an increased risk of being diagnosed with incident heart failure (1.6 fold), diabetes mellitus (1.6 fold), and malnutrition (1.4 fold) than patients with normal BMI, while obese patients showed an increased risk of being diagnosed with incident diabetes mellitus (1.8 fold) and sleep disorder (1.3 fold). However, if we conducted an association analysis between BMI and any PASC conditions, we found you mentioned pattern underweight (aHR 1.17 (1.12-1.23)) and obesity (aHR 0.98 (0.93-1.02)) after fully adjustment. I think this highlights some strengths of our analysis which is trying to show the associations of these covariates and a range of heterogeneous PASC conditions.

Minor comments:

- The title should be revised. "Predictive modeling" is an action phrase, so the first phrase should match. Perhaps "Identifying risk factors and predictive modeling of post-acute ..."?

Response: Thanks for the comments. We have revised the title as suggested.

- Line 76, for citations 3 and 5 both, page 19 is referenced. Citation 3 has page numbers that don't include 19 (they range from 259 to 264), and citation 5 is a 13-page document. Please correct citations.

Response: Thanks for the comments. We have updated these citations.

- Line 89, the phrase "systematic study" seems like puffery. "Study" should suffice.

Response: Thanks for the comments. We have replaced the “systematic” with “data-driven”.

- Line 93, “used as” should be changed to “used for”.

Response: Thanks for the comments. We have revised it as suggested.

- Citation 14 should include “medRxiv” in it.

Response: Thanks for the comments. We have replaced this medRxiv citation with its published nature communications version.

- Line 115, “real-world” seems like puffery. EHR data is typically from real world. There is no need to emphasize that.

Response: Thanks for the comments. We have deleted it as suggested.

- The first table in the Supplement (pages 1-138 of the supplement) appears to have a lot of duplicative rows. Was it exported correctly? Should it be de-duplicated? Looking at pages 277 and beyond, I wonder if this was supposed to be seen as a table with five columns, but the way it was exported makes it difficult to align the cells that are on the same row. Can it be re-exported, perhaps in a “wide” format (landscape, as opposed to portrait)?

Response: Thanks for the comments. We would like to do the following clarifications. The Supplementary code list has unique rows regarding their ICD-10-CM Code column (column D). Multiple rows/ICD codes can be assigned into the same CCSR Category (column B).

- Line 230, the word “in” is missing before “a predefined hyperparameter space”.

Response: Thanks for the suggestions. We have revised this sentence.

- Line 250 mentions the first “3-digits” of ICD-10 codes. ICD-10 codes are not numbers; they in fact start with a letter. Either authors mean the “first 3 characters”, or they may “the first 4 characters” (letter + 3 digits). This should be clarified. Moreover, ICD-10 hierarchy doesn’t always work this way, i.e., the first 3 characters are not always going to give you a roll up of equal granularity. Authors should justify their choice.

Response: Thanks for the comments. We would like to do the following clarifications.

In our primary analysis we used clinician-selected predictors. Here, we conducted a sensitivity analysis on how different feature engineering method will impact the conclusion of predictability of different PASC conditions. Here we used a data-driven way by aggregating different ICD codes by their first 3 characters and medications by their RxNorm-CUI ingredients. This aggregation is one of the most widely used methods in predictive modeling work. Indeed, more fine-grained roll-up method can be applied to ICD codes.

- Line 284, “has” should be changed to “have”.

Response: Thanks for the comments. We have revised this prediction performance section.

- Lines 321-322 include an incomplete sentence.

Response: Thanks for the comments. We have revised it as “Here we analyzed the associations between the covariates and the risk of developing any incident PASC conditions, quantified by the unadjusted hazard ratio (HR) and fully adjusted hazard ratio (aHR).”

- The GitHub link on lines 554-555 does not work.

Response: Thanks for the comments. We have updated the link and kindly check https://github.com/calvin-zcx/pasc_phenotype.

Reference

1. Zang, C. *et al.* Data-driven analysis to understand long COVID using electronic health records from the RECOVER initiative. *Nat. Commun.* **14**, 1948 (2023).
2. Zhang, H. *et al.* Data-driven identification of post-acute SARS-CoV-2 infection subphenotypes. *Nat. Med.* 1–10 (2022) doi:10.1038/s41591-022-02116-3.
3. COVID-19 and iron deficiency anemia: relationships of pathogenesis and therapy | Gromova | Obstetrics, Gynecology and Reproduction.
https://www.gynecology.su/jour/article/view/831?locale=en_US.

4. Khullar, D. *et al.* Racial/Ethnic Disparities in Post-acute Sequelae of SARS-CoV-2 Infection in New York: an EHR-Based Cohort Study from the RECOVER Program. *J. Gen. Intern. Med.* (2023) doi:10.1007/s11606-022-07997-1.
5. Thaweethai, T. *et al.* Development of a Definition of Postacute Sequelae of SARS-CoV-2 Infection. *JAMA* (2023) doi:10.1001/jama.2023.8823.
6. Post-COVID-19 conditions among adult COVID-19 survivors aged 18–64 and ≥65 years — Cerner Real World Data, United States, March 2020–November 2021. <https://stacks.cdc.gov/view/cdc/117411>.

Reviewers' comments:

Reviewer #2 (Remarks to the Author):

The authors have addressed several concerns in their revised manuscript, though factors to address remain.

1. Regarding vaccine status, it is interesting that so few (1%) individuals had a vaccine prior to their first Covid infection. Is this a reflection of missing vaccine data? If not, I would be concerned about a sampling error.
2. Regarding discharge location, the previously posed question was referring where the patients are discharged too. For example, did they return home or did they go to a long term care facility or nursing home after their acute care hospitalization? There are likely differences in the rate of pressure ulcers, for example, among patients in nursing homes than those at home. It is not clinically appropriate to attribute a new diagnosis of a pressure ulcer to a hospitalization that occurred more than 30 days prior.
3. I would recommend removing smoking categorization from the methods section sense it has been removed from the analysis.
4. Was there a reason to categorize age and ADI rather than leaving as continuous variables?

My overall largest concern with this manuscript reflects likely spurious associations, despite Bonferroni correction. Further, the lack of granularity in the severity of illness variables is limiting. These limitations make interpretation of the results difficult at best and clinically less informative.

Reviewer #3 (Remarks to the Author):

Authors have addressed my prior comments reasonably and sufficiently. I don't have any further comments at this time. The decision to publish should be based on the scientific value of the study. This study is observational in nature and subject to limitations of any study that is primarily based on ICD codes, so it is certainly susceptible to confounding.

Reviewer #4 (Remarks to the Author):

In general, the authors of this paper seem to have followed a valid approach and (mostly) outline their approach and limitations clearly. I think with a few tweaks this can be published.

The one area of reviewer 1's comments that I thought the authors could respond more clearly to is the last paragraph of reviewer 1's comments (about providing more detail on the predictive models and whether those would be clinically useful for identifying at-risk individuals). I think the paper could still include some more detail about this aspect.

I also had a look at the other reviewer comments and the authors' responses. In general, I think that the authors have responded well, but there are a number of items which I think the authors ought to

highlight in the paper itself, rather than just in the response to reviewers. Since there are quite a few of these, I'll attach my annotated .pdf document so you can take a look yourself at my highlights and comments.

Referee expertise:

Referee #1: MD in Infectious Disease, epidemiology

Referee #2: MD in cardiology, healthcare technology, electronic health records

Referee #3: MD, biomedical informatics

Reviewers' comments:

We appreciate the thoughtful comments from the reviewers and provide our point-by-point response in this revision as below.

Reviewer #1 (Remarks to the Author):

"Risk factors and predictive modeling for post-acute sequelae of SARS-CoV-2 Infection: Findings from EHR Cohorts of the RECOVER Initiative" is a retrospective cohort study involving two large datasets that aims to first identify risk factors for incident PASC conditions and then to develop models to predict who will develop these conditions. Although the authors use several advanced analytic models, **the method of cohort creation is fundamentally flawed and there is limited adjustment for case-ascertainment bias, which likely explains the vast majority of findings, which are more likely to be artifact due to new and increased engagement in care than true findings.** Several of the associations are quite clearly spurious (e.g., COVID as the cause of anemia during pregnancy) and suggest additional attention to potential for cause and effect is needed before moving forward. **Additional clinical engagement about natural history of disease and causal pathways is strongly encouraged.**

As noted above, the study is heavily influenced by **case ascertainment bias and regular versus non-regular users of care. Although the number of visits prior to COVID diagnosis is measured, it is not accounted for.** The importance of regular engagement of the healthcare system as a driver of findings is evident from table 1. Compared to those with COVID infections, those with PASC had fewer inpatient visits, outpatient visits, and emergency room visits during the three-year period prior to COVID diagnosis. They also had lower rates of baseline morbidities AND higher median area deprivation indices, which additionally

suggests that **most of the incident codes are due to new engagement with the healthcare system and not attributable to COVID-19**. Methodologically, this could be addressed by matching on prior engagement with the healthcare system based on number of prior visits, rather than requiring ICD one code (without even a visit required) to measure "incident cases of PASC." (This same methodologic error is present in many of the PASC studies, including those that are heavily referenced in the paper).

Response: Thanks for the comments. We would like to do the following clarifications.

Regarding the engagement concerns, first, we not only measured but also adjusted for the history of healthcare engagement covariates in our association analysis (Methods-covariates and we further clarified these in the discussion section). Second, Regarding incident PASC codes, in our previous literature, we also adopted IPTW ¹ and matching ² to account for the engagement. Third, compared with existing association analysis literature, we further compared the non-infected group and required the identified association should be enhanced compared with the non-infected group. Because the bias of the new engagement argument can also hold for non-infected groups and our approach might mitigate your proposed concerns (Methods-Association analysis-step II).

Regarding the potentially spurious associations. One example was baseline pregnancy and anemia, as anemia is the most common hematologic problem in pregnancy. However, there were also studies suggesting that SARS-CoV-2 infection during pregnancy can further exacerbate iron deficiency anemia due to hyperinflammation during the acute infection phase³ (in our discussion section). What distinguishes our work from existing association literature is our efforts to get rid of these spurious associations. First, we required additional/marginal association compared with non-infected patients (Method-association analysis section). Also taking anemia in pregnancy as an example, we observed worsened associations than the non-infected control group (Fig 1) and we removed a lot of potentially spurious associations as shown in the sensitivity analysis (Extended Fig 1). Second, we also used the corrected P-value (<0.000562) to identify likely associations, aiming to adjust false/spurious findings in the multiple test settings (Method-association analysis section). We identified our work as an association hypothesis generation work, which can facilitate further mechanism studies to confirm the risk factors.

In addition, the authors identify risk factors for severe COVID-19 (e.g., COPD, dementia, ICU, older age) and it is not clear the extent to which these identified associated are due to PASC or due to risk of severe disease \diamond healthcare engagement/ICU admission \diamond new case identification. Complete re-analysis with improved adjustment for healthcare engagement and timing of new diagnosis would be necessary to address this major limitation. Two articles that describe risk factors for severe disease despite vaccination are:

[https://www.thelancet.com/journals/lancet/article/PIIS0140-6736\(22\)01656-7/fulltext](https://www.thelancet.com/journals/lancet/article/PIIS0140-6736(22)01656-7/fulltext)
<https://jamanetwork.com/journals/jamanetworkopen/fullarticle/2797495>

Response: Thanks for the comments. We would like to do the following clarifications.

We also hypothesized that the severity during acute infection were factors driving PASC risks. First, we tried to approximate acute severity by their hospitalization, ICU, and ventilation status (1 day before to 16 days after the acute infection) and quantified their associations with a range of incidence conditions in the post-acute phase (30 days to 180 days after the infection). As shown in Fig1, the severe acute infection was associated with incident COPD and dementia. Second, we further conducted a subgroup analysis by stratifying patients by their severity in the acute infection phase (hospitalized or non-hospitalized) and then performing statistical analysis within each stratum. The noninfected control patients were also stratified according to the hospitalized or non-hospitalized during the acute period, to capture background associations within each subgroup population. We can see different association patterns across both two groups, as summarized in Fig 3, and Associations stratified by the acute care settings section. We found acute ICU status was also associated with incident post-acute COPD condition.

As we have replied in our previous response, we have adjusted for baseline healthcare engagement in our analysis, and further using non-infected control patients and corrected P-values to eliminate the concerns of spurious findings.

Some specific examples of likely spurious associations for attribution to COVID/PASC (there are many, this is not an exhaustive list):

- Diagnosis of anemia following COVID diagnosis in pregnant patients (this is likely due to new engagement in care, and the expected outcome of pregnancy, which is birth) (we required riskier than control group)
- Association between low BMI, COVID, and malnutrition. The malnutrition diagnosis is almost certainly secondary to the low BMI, not the incident COVID diagnosis.
- Association between cancer and malnutrition (cancer and chemotherapy cause malnutrition)
- Association between Parkinson's disease and encephalopathy
- CKD and heart failure/anemia (CKD causes anemia)
- Association between obesity and diabetes (again, obesity causes diabetes)

Given the fundamental methodologic concerns noted above, I will not make specific comments about the tables, figures, etc, as these would require complete revision after re-analysis in order for this article to be suitable for publication.

Response: Thanks for the comments. We have detailed our methods for reducing likely spurious associations in our first response, briefly including additional/marginal association compared with non-infected patients, fully adjustment for baseline covariates (including the mentioned healthcare utilization), and corrected P-values for multiple testing. The above-mentioned associations were

worsened due to covid infection compared with associations in non-infected patients by following our approach. For associations not exhibiting worsening associations were summarized in Extended data fig 1.

However, as structured, several sections of the manuscript are quite repetitive (E.g., methods and results section contain close to the same paragraph) and the manuscript as a whole could be reduced by ~25% without losing any content. Additionally, the major focus of the results section is on the associations listed above, rather than the predictive models. After reading the results section, it is not clear what is included in the models and whether those models would be clinically useful tools for identifying risk. Strongly suggest removing the redundancy and including more detail about model findings and how they could be used to inform clinical practice.

Response: Thanks for the comments. We have revised the method and results sections as suggested.

Reviewer #2 (Remarks to the Author):

In their manuscript, "Risk Factors and Predictive Modeling for Post-Acute Sequelae of SARS-CoV-2 Infection: Findings from EHR Cohorts of the RECOVER Initiative," Zang et al describe associations between EHR derived conditions and the risk of various PACS-conditions/symptoms. The authors should be applauded for tackling such a complex issue and attempting to do so using EHR data, which tends to lend itself to broader use once validated. There are several limitations which warrant further evaluation. In particular, despite excellent efforts to control for it, residual confounding likely remains, driven in part by changes in engagement with the healthcare system following SARS-CoV-2 infection. Further, the outcomes require additional clinical context for, as the authors note, several of the associations are among conditions that are themselves highly correlated with each other.

1. Number of patients listed in abstract is different than the numbers listed in lines 117-118. Further, saying that there were 12 and 15 million patients in each cohort is misleading as there were far less in the final cohorts, as indicated in Figure 1.

Response: Thanks for the comments. We have revised our abstract. Also, we would like to do the following clarifications. These are the base population in our two Clinical Research Networks and we described these base populations in the method section (also in the top rectangles in Fig1). As you mentioned, the final selected cohorts (the bottom rectangles in Fig1) were much smaller and we explicitly show the selection details in the results section and Fig 1.

2. Given the degree of missingness for smoking (90.2%) noted on line 184, is it worth even including this variable in the model?

Response: Thanks for the comments. We have revised our primary analyses by excluding the smoking variables and we also acknowledged this limitation due to the missingness in the discussion section.

3. How was death within 30 days of diagnosis handled?

Response: Thanks for your questions. We would like to do the following clarifications. Because of the intrinsic definition of the post-acute sequelae of SARS-CoV-2 infection (PASC), we selected patients who had at least one diagnosis code from 31 days to 180 days after the index date (referred to as the post-acute phase or follow-up period), to ensure that patients were connected to the healthcare system, were being observed during the study period, and were being alive beyond the acute phase. This is a criterion adopted in RECOVER initiative to screen the PASC signals.^{1,2,4} Thus, the patients who died in the acute phase were typically not included in the PASC/Long Covid studies.

4. Several other important variables should be considered including the repeat/multiple infections, vaccination status (mentioned in the discussion, but a major limitation given data indicating that this may reduce severity of illness and PASC risk), and treatment with SARS-CoV-2 specific therapies.

Response: Thanks for your questions. We would like to do the following clarifications. First, we didn't use vaccination status because a) the vaccine began in early December 2020 and more than half of the patients got infected before the vaccine was available. Even for patients after December 2020, the number of recorded patients was limited, please see the table below. We defined the fully vaccinated as two shots of mRNA vaccine (Pfizer, or Moderna) or one shot of J&J, see <https://www.cdc.gov/coronavirus/2019-ncov/vaccines/stay-up-to-date.html>. In all, the vaccine baseline covariates before the index date only accounted for 2% of the population in the covid+ group and 5% in the covid- group, barely changing the final screened conditions considering their significant hazard ratio. We acknowledged this as a limitation in the discussion section.

	INSIGHT COVID+		INSIGHT COVID-	
	N	%	N	%
Total number	61305	100%	577174	100%
Fully vaccinated - Pre-index	811	1%	17229	3%
Partially vaccinated - Pre-index	823	1%	11753	2%
No evidence - Pre-index	59671	97%	548197	95%

Second, Studying how the vaccine influences the risk of long Covid is a huge topic by itself and left as a future study with a more sophisticated experiment design, and also relies on the ongoing efforts of cumulating and collecting more vaccination data (e.g. registry database, which is not available yet).

5. Is the discharge location of patients known? This is of particular importance in the development of conditions such as pressure ulcers, which are more likely to occur in individuals living in care facilities.

Response: Thanks for your comments. As shown in Fig 3 (subgroup/stratified analysis by the hospitalization status), we investigated factors associated with incident PASC when we stratified patients according to their hospitalization status during their acute infection. We observed pressure ulcers were associated with ICU, ≥ 75 ages, weight loss at baseline for hospitalized patients only, but not for the non-hospitalized patients. These findings were aligned with your proposed comments that pressure ulcers were more likely to occur in individuals living in care facilities.

6. Severity of illness is clearly a driving factor for PACS risk, particularly myopathies, pressure ulcers, malnutrition, and U099/B948 codes. With the exception of the latter, however, ICU stays (particularly long stays) place individuals at increased risk for these conditions regardless of Covid status. While the authors worked to control for this with their non-infected group, the lack of granularity in the medical use variable may overlook differences in true severity of illness. For example, a Covid patient who spent a month in the ICU on a ventilator would be considered as having the same severity of illness as a non-Covid patient who spend a night in the ICU after an elective surgery. Was consideration given to using a more granular variable, such as the WHO ordinal clinical severity scale?

Response: Thanks for your comments. We would like to do the following clarifications. First, we use hospitalization status, ICU, and ventilation status during the 1 day prior to 16 days after their index date to capture the severity of illness during the acute phase. See a copy of top rows of table 1 below. These modeling of acute severity were consistent with our previous literature^{1,2,4,5} and CDC work⁶.

Characteristics	Number of Patients with Lab-Confirmed SARS-CoV-2 Infection (columns %)	Number of Patients with Lab-Confirmed SARS-CoV-2 Infection and with at least One Incident PASC Condition (columns %)
Total	35,275	17,571 (49.8)
The Severity of Acute Infection — no. (%)		
Not hospitalized	22,148 (62.8)	9,809 (55.8)
Hospitalized w/o ICU	11,480 (32.5)	6,611 (37.6)
ICU	1,647 (4.7)	1,151 (6.6)

Second, we conducted a stratified analysis by stratifying patients by their severity in the acute infection phase (hospitalized or non-hospitalized) and then performing statistical analysis within each stratum. The noninfected control patients were also stratified according to the hospitalized or non-hospitalized during the acute period, to capture background associations within each subgroup population. We can see different association patterns across both two groups, as summarized in Fig 3, and Associations stratified by the acute care settings section. Third, we acknowledge potential granularity issues of acute severity and a small number of ICU patients (a limited number of Covid patients who spent a month in the ICU) in the discussion section and we would like to explore more fine-grained severity scales, like the WHO ordinal clinical severity scale, in our future analysis as more patients were accumulated.

7. Lines 289-306 are redundant from the methods and should be considered for removal or merger with the methods section.

Response: Thanks for your comments. We have removed redundant content and systematically reorganized the method and results sections.

8. In Table 1, calculation of percentages of the total for number of patients with Covid and percentages of the stratum among those with PASC makes interpretation of the true prevalence of these conditions in those with PACS difficult. Consider calculating both as percent of total for each column.

Response: Thanks for your suggestions. We have revised the table 1 by calculating percentage within each column.

9. In Extended Table 1, the HR for PACS declines with increasing number of outpatient visits prior to infection. Did the authors consider that this may be a reflection that patients with greater engagement with the healthcare system prior to infection received diagnoses before infection (either resulting in or because of greater engagement with healthcare providers); conversely, patients who had limited healthcare engagement prior to Covid infection subsequently had greater opportunity to be diagnosed with new conditions, simply because of greater engagement?

Response: Thanks for your comments. We would like to do the following clarifications. It's possible, and thus, we tried to screen likely associations by a) adjusting for a range of baseline covariates including these baseline hospitalization utilization features, and b) further compared these associations in both infected patients versus non-infected patients. The same arguments can be applied to the non-infected patients and by requiring excess risk over this control group can further mitigate the proposed concern.

10. Please clarify the methods for the sensitivity analysis using a 'data-drive approach.' The authors state that they used the first 3 digits of ICD-10 codes and medications codes. It is unclear what was actually done and how this is of benefit over a theory driven approach.

Response: Thanks for your comments. We would like to do the following clarifications. In our primary analyses, we used a list of clinician-selected baseline clinical features^{1,2,4} (theory-driven approach, relatively low dimensional) for association analysis and predictive modeling. In addition, we also check more high-dimensional baseline covariates by considering all the recorded ICD-10 codes (aggregated by their first 3-characters) and prescriptions (aggregated by their active ingredients). The use of different feature engineering methods is aimed to validate the robustness of the predictability of PASC.

Reviewer #3 (Remarks to the Author):

Authors used data from two large PCORNet data sets to explore which risk factors may be associated with the occurrence of post-acute sequelae SARS-CoV-2 infection (PASC).

Major comments:

- Since this study is observational in nature, causal inferences should not be implied. Several phrases need to be edited to that end. For example, line 105 "will likely improve" implies causal inference. Similar edits need to happen in the last paragraph of Discussion.

Response: Thanks for the comments. We have deleted this implication in line 105 and revised the discussion section accordingly.

- To define PASC, authors use the incidence of a diagnosis from a long list of conditions (from depression and anxiety to myopathies and acute kidney failure) from 31 to 180 days after COVID infection. This will likely include a lot of false positives (people can have those conditions, regardless of COVID too). They capture all of these using ICD-10 codes. The only ICD-10 codes that are specific (though not necessarily sensitive) for PASC are U099 and perhaps B948. Authors should do a sensitivity analysis, in which they only use these two diagnoses to capture the outcome of PASC. This is important because some of the risk factors they are claiming to be related to PASC (such as obesity) are indeed risk factors for COVID-unrelated incidence of those conditions in the long list, so using a narrow, more specific definition for PASC may validate (or refute) if these risk factors are indeed related to PASC.

Response: Thanks for your comments. We would like to do the following clarifications. First, we have used the general PASC codes U099/B948 as one outcome in our PASC list and we have reported the association results regarding U099/B948 codes in our results section. As shown in Figure 1, patients with weight loss or obesity (BMI ≥ 30), or who got hospitalized or in ICU during their acute period were at a higher risk of being diagnosed with these general PASC codes. Second, the U099 codes were effective since 10/1/2021, which can not capture patients who got infected before that. Third, defining PASC is still an open question and our PASC conditions were consistent with our previous literature^{1,2,4,5}, which usually exhibited excess burden over the non-infected control group. Regarding association analyses, to reduce the change of false findings, we first adopted non-infected control groups and required the adjusted hazard ratio value of the identified association estimated from the infected patients to be larger than the value estimated from non-infected patients, and we further adopted very stringent significance level corrected due to the multiple test settings.

- Authors do try to address the above concern using a second Cox model that include all patients. However, in lines 216-221 they only require the direct association to be statistically significant, and they don't seem to enforce statistical significance for the confirmatory interaction term in the second model. The results of this model with interaction terms are also not fully reported; I strongly suggest including that as a table in the supplement. In fact, one could argue that the risks reported in lines 337-397 should be replaced with those calculated on the interaction terms in the second model. Right now, it seems like these risks are from the first model, so the fact that obese patients had a 1.4-fold increased risk of

diabetes may not describe anything about COVID-related diabetes and be merely a representation of the (well known) association between obesity and diabetes.

Response: Thanks for your comments. We have reported new results of aHR of the interaction term in Extended Data Fig 2 (compared with primary results in Fig 2), and conditions if we further required interaction terms to be significant in Extended Data Fig 3, and reported these new results in the sensitivity analysis section. Existing literature majorly reported their results based on model 1 (Level 1), and we required more stringent screening criteria in our analysis (aHR in model 1, $p\text{Value} < 0.000562$ under multiple test setting, and aHR in model 2 > 1 , denoted as Level 2). We reported our primary results in Fig 2 and the previous method in Extended Data Fig 1. as a sensitivity analysis. In addition, we further reported results if pValue of aHR in model 2 is significant in Extended Data Fig 3 as another sensitivity analysis (Level 3). By using more and more stringent criteria, we can expect potentially fewer identified associations in Fig 2 than in Extended Data Fig 1, and fewer associations in Extended Data Fig 3 than in Fig 2. This is a trade-off between sensitivity and precision, and because current knowledge is based on Level 1, and thus we reported Level 2 as a primary analysis and Level 3 as a sensitivity analysis. We identified our work as a data-driven association analysis/hypotheses generation work, which can be helpful for further mechanism study to confirm risk factors for the long covid.

- On line 200 authors state that “loss of follow-up” is used as a censoring event. However, their cohort, by definition, only includes patients who were followed up (because cohort definition requires at least one diagnosis code in days 31 to 180 after COVID incidence, which is defined as “follow up period” in line 198). Authors need to explain how they are defining loss of follow-up.

Response: Thanks for the comments. We would like to do the following clarifications. First, we required any diagnosis in days between 31 to 180 to require that a patient was alive beyond the acute phase, and he/she was connected to the EHR system and got the health status captured. This is a criterion adopted in RECOVER initiative to screen the PASC signals.

Second, the loss of follow-up in the database is used to define a censoring event. “The censoring event is defined as the earliest event of documented death, loss of follow-up in the database (the date of the last documented records in the EHR systems), 180 days after the baseline, or the end of our observational window (December 31, 2021).” For example, if a patient’s records ended at, say day 90, without documented PASC conditions, we would say that this patient lost follow-up at day 90 in the EHR databases, namely event time was ≥ 90 .

Third, we added a short description “(the date of the last documented records in the EHR systems)” to further explain the loss of follow-up in the database.

- Lines 298-309 really belong in Methods, not results. Lines 322-328 are repeating what was already stated in Methods.

Response: Thanks for your great comments. We have reorganized the result section and removed duplicated method contents.

- More inpatient or outpatient visits in the past 3 years seem to be negatively associated with PASC. How do authors explain this?

Response: Thanks for the comments. We would like to do the following clarifications. First, we didn't find significant associations between past healthcare utilization and a list PASC conditions as shown in Fig 2 (the association map in our primary results, which were fully adjusted, background signal removed, and significant under multiple tests). Second, the associations of past healthcare utilization and having at least one PASC condition were summarized in Table 2. In terms of univariate analysis, we found slightly increased HR as the number of past inpatient visits increase. However, when adjusted for all the covariates we found you mentioned "negative" associations. A possible reason is we accounted for other baseline covariates including a list of comorbidities for which the hospital visits were just proxy. Thus, we adopted very stringent screen criteria/methods to identify more likely/direct risk associations and reported them in Fig 2.

Covariate	Univariate HR (95% CI)	Univariate - log10 (P-Value)	Fully adjusted HR (95% CI)	Fully adjusted - log10 (P-Value)
No. of hospital visits in the past 3 yrs				
Inpatient 0	1.00 (ref)		1.00 (ref)	
Inpatient 1-2	1.16 (1.12-1.20)	15.3	0.96 (0.92-1.00)	1.3
Inpatient ≥ 3	1.17 (1.11-1.23)	8.6	0.87 (0.82-0.93)	4.1
Outpatient 0	1.00 (ref)		1.00 (ref)	
Outpatient 1-2	0.71 (0.66-0.76)	20.3	0.80 (0.74-0.86)	9
Outpatient ≥ 3	0.62 (0.58-0.66)	57.7	0.64 (0.60-0.68)	47.1

- Later infection periods (after 3/21) seem to be positively associated with PASC. How do authors explain this?

Response: Thanks for the comments. We would like to do the following clarifications. We have reported the associations of later infection periods (compared with the early period 3/20 to 6/20 as reference) in the result section Period of infection. This later period was dominated by the Delta variant of SARS-CoV-2, and as shown in our primary results Fig 2, we observed an increased risk of being diagnosed with incident pharyngitis (3.2-fold), chest pain (1.9-fold), abdominal pain (1.7-fold), dyspnea (1.6-fold), as well as being diagnosed with general PASC symptoms and signs with the U099/B948 ICD codes (5-fold) in the post-acute infection period compared to patients got infected during the 1st wave as the reference period. We think these results implied the potentially different PASC manifestations and associations for different variants of concerns over different periods.

- Prior research has repeatedly suggested that obesity is associated with PASC. In this study, it doesn't appear associated, and in fact, being underweight is associated with PASC. How do authors explain this?

Response: Thanks for the comments. We would like to do the following clarifications. We discussed the associations between BMI and a range of PASC conditions in the result section Body Mass Index. As shown in our primary result in Fig 2, we found BOTH obese and underweight were at higher risk of being diagnosed with a certain but different set of PASC conditions. Specifically, underweight patients were at an increased risk of being diagnosed with incident heart failure (1.6 fold), diabetes mellitus (1.6 fold), and malnutrition (1.4 fold) than patients with normal BMI, while obese patients showed an increased risk of being diagnosed with incident diabetes mellitus (1.8 fold) and sleep disorder (1.3 fold). However, if we conducted an association analysis between BMI and any PASC conditions, we found you mentioned pattern underweight (aHR 1.17 (1.12-1.23)) and obesity (aHR 0.98 (0.93-1.02)) after fully adjustment. I think this highlights some strengths of our analysis which is trying to show the associations of these covariates and a range of heterogeneous PASC conditions.

Minor comments:

- The title should be revised. "Predictive modeling" is an action phrase, so the first phrase should match. Perhaps "Identifying risk factors and predictive modeling of post-acute ..."?

Response: Thanks for the comments. We have revised the title as suggested.

- Line 76, for citations 3 and 5 both, page 19 is referenced. Citation 3 has page numbers that don't include 19 (they range from 259 to 264), and citation 5 is a 13-page document. Please correct citations.

Response: Thanks for the comments. We have updated these citations.

- Line 89, the phrase "systematic study" seems like puffery. "Study" should suffice.

Response: Thanks for the comments. We have replaced the “systematic” with “data-driven”.

- Line 93, “used as” should be changed to “used for”.

Response: Thanks for the comments. We have revised it as suggested.

- Citation 14 should include “medRxiv” in it.

Response: Thanks for the comments. We have replaced this medRxiv citation with its published nature communications version.

- Line 115, “real-world” seems like puffery. EHR data is typically from real world. There is no need to emphasize that.

Response: Thanks for the comments. We have deleted it as suggested.

- The first table in the Supplement (pages 1-138 of the supplement) appears to have a lot of duplicative rows. Was it exported correctly? Should it be de-duplicated? Looking at pages 277 and beyond, I wonder if this was supposed to be seen as a table with five columns, but the way it was exported makes it difficult to align the cells that are on the same row. Can it be re-exported, perhaps in a “wide” format (landscape, as opposed to portrait)?

Response: Thanks for the comments. We would like to do the following clarifications. The Supplementary code list has unique rows regarding their ICD-10-CM Code column (column D). Multiple rows/ICD codes can be assigned into the same CCSR Category (column B).

- Line 230, the word “in” is missing before “a predefined hyperparameter space”.

Response: Thanks for the suggestions. We have revised this sentence.

- Line 250 mentions the first “3-digits” of ICD-10 codes. ICD-10 codes are not numbers; they in fact start with a letter. Either authors mean the “first 3 characters”, or they may “the first 4 characters” (letter + 3 digits). This should be clarified. Moreover, ICD-10 hierarchy doesn’t always work this way, i.e., the first 3 characters are not always going to give you a roll up of equal granularity. Authors should justify their choice.

Response: Thanks for the comments. We would like to do the following clarifications.

In our primary analysis we used clinician-selected predictors. Here, we conducted a sensitivity analysis on how different feature engineering method will impact the conclusion of predictability of different PASC conditions. Here we used a data-driven way by aggregating different ICD codes by their first 3 characters and medications by their RxNorm-CUI ingredients. This aggregation is one of the most widely used methods in predictive modeling work. Indeed, more fine-grained roll-up method can be applied to ICD codes.

- Line 284, “has” should be changed to “have”.

Response: Thanks for the comments. We have revised this prediction performance section.

- Lines 321-322 include an incomplete sentence.

Response: Thanks for the comments. We have revised it as “Here we analyzed the associations between the covariates and the risk of developing any incident PASC conditions, quantified by the unadjusted hazard ratio (HR) and fully adjusted hazard ratio (aHR).”

- The GitHub link on lines 554-555 does not work.

Response: Thanks for the comments. We have updated the link and kindly check https://github.com/calvin-zcx/pasc_phenotype.

Reference

1. Zang, C. *et al.* Data-driven analysis to understand long COVID using electronic health records from the RECOVER initiative. *Nat. Commun.* **14**, 1948 (2023).
2. Zhang, H. *et al.* Data-driven identification of post-acute SARS-CoV-2 infection subphenotypes. *Nat. Med.* 1–10 (2022) doi:10.1038/s41591-022-02116-3.
3. COVID-19 and iron deficiency anemia: relationships of pathogenesis and therapy | Gromova | Obstetrics, Gynecology and Reproduction.
https://www.gynecology.su/jour/article/view/831?locale=en_US.

4. Khullar, D. *et al.* Racial/Ethnic Disparities in Post-acute Sequelae of SARS-CoV-2 Infection in New York: an EHR-Based Cohort Study from the RECOVER Program. *J. Gen. Intern. Med.* (2023) doi:10.1007/s11606-022-07997-1.
5. Thaweethai, T. *et al.* Development of a Definition of Postacute Sequelae of SARS-CoV-2 Infection. *JAMA* (2023) doi:10.1001/jama.2023.8823.
6. Post-COVID-19 conditions among adult COVID-19 survivors aged 18–64 and ≥65 years — Cerner Real World Data, United States, March 2020–November 2021. <https://stacks.cdc.gov/view/cdc/117411>.

RESPONSE TO REVIEWER COMMENTS

We highly appreciate the careful, critical, and constructive comments from all the reviewers. We have revised the manuscript substantially in response to the comments. In the following, we provide a detailed point-by-point response to these comments in this revision. We hope you will find these revisions satisfactory.

Reviewers' comments:

Reviewer #2 (Remarks to the Author):

The authors have addressed several concerns in their revised manuscript, though factors to address remain.

1. Regarding vaccine status, it is interesting that so few (1%) individuals had a vaccine prior to their first Covid infection. Is this a reflection of missing vaccine data? If not, I would be concerned about a sampling error.

Response: Thanks for your questions. We would like to make the following clarifications. Several factors contribute to this low ratio: First, nearly half of the study population got infected before any vaccine was available, as shown in the figure below:

Supplementary Fig. 1. Temporal trends of the lab-confirmed monthly new SARS-CoV-2 cases per 10,000 patients in the INSIGHT and OneFlorida+ cohorts, March 2020 to November 2021.

Ref: https://static-content.springer.com/esm/art%3A10.1038%2Fs41467-023-37653-z/MediaObjects/41467_2023_37653_MOESM1_ESM.pdf

Second, indeed, regarding the remaining half of the study population, vaccine records collected not in the hospitals were missing (e.g., registry database, which is not available yet) for the general population, and we have acknowledged this as a limitation in the discussion section and treated the linkage to the vaccine data as one of the major tasks in the future RECOVER efforts.

2. Regarding discharge location, the previously posed question was referring where the patients are discharged too. For example, did they return home or did they go to a long term care facility or nursing home after their acute care hospitalization? There are likely differences in the rate of pressure ulcers, for example, among patients in nursing homes than those at home. It is not clinically appropriate to attribute a new diagnosis of a pressure ulcer to a hospitalization that occurred more than 30 days prior.

Response: Thanks for your great suggestion. Indeed, pressure ulcers are more likely to occur in hospitalized patients or individuals living in care facilities. We have excluded pressure ulcers from our likely PASC list and updated all the primary results in Fig 2-4 and extended results in extended data fig 1-5.

3. I would recommend removing smoking categorization from the methods section sense it has been removed from the analysis.

Response: Thanks for your suggestion. We have removed the smoking-related descriptions from the method section.

4. Was there a reason to categorize age and ADI rather than leaving as continuous variables?

Response: Thanks for your questions. We would like to make the following clarifications. The ADI is a discrete ranking number starting from 1, 2, to 100. We categorized these 100 dimensions into groups with different levels of disadvantage to reduce the dimension of covariates in a similar spirit or purpose to categorizing age or BMI into age groups or BMI groups, and also try to be consistent with our previous analyses.^{1,2}

My overall largest concern with this manuscript reflects likely spurious associations, despite Bonferroni correction. Further, the lack of granularity in the severity of illness variables is limiting. These limitations make interpretation of the results difficult at best and clinically less informative.

Response: Thanks for your great comments. We would like to make the following clarifications. First, we have substantially updated our primary results in Fig 2-4 and highlighted our primary findings and potential implications for the clinical practice. To reduce potential spurious associations, we have further highlighted a smaller list of associations under more strict screening criteria (Fig 2-4 and associated results section). Second, we further summarize our interpretations of the results as follows (as well as in the discussion section):

1. We found complex association patterns and varying predictability of incident PASC conditions which may represent a challenge for managing heterogeneous PASC conditions.
2. Thus, we suggest using machine learning-based predictive models and HER, through managing the abovementioned complexity, could help in identifying patients who were at risk of developing different incident PASC conditions.
3. Among complex association patterns, we further highlight severe acute infections, being underweight, and having baseline conditions including cancer or cirrhosis that are potentially associated with overall incident PASC in the post-acute phase
4. We suggest further investigation of the association between COVID-19 treatment in adults who are at high risk for severe COVID-19 and the risk of PASC beyond the acute phase (bearing the hypothesis that preventing severe acute infection might be associated with a lower risk of incident long-term conditions).

Third, we have discussed more on the granularity of the severity in the discussion section, and regard the modeling of the ICU/Hospitalization stay or WHO ordinal clinical severity scale as our future focus as more ICU/hospitalized patients accumulate over time.

Reviewer #3 (Remarks to the Author):

Authors have addressed my prior comments reasonably and sufficiently. I don't have any further comments at this time. The decision to publish should be based on the scientific value of the study. This study is observational in nature and subject to limitations of any study that is primarily based on ICD codes, so it is certainly susceptible to confounding.

Response: Thanks again for your constructive comments.

Reviewer #4 (Remarks to the Author):

In general, the authors of this paper seem to have followed a valid approach and (mostly) outline their approach and limitations clearly. I think with a few tweaks this can be published.

The one area of reviewer 1's comments that I thought the authors could respond more clearly to is the last paragraph of reviewer 1's comments (about providing more detail on the predictive models and whether those would be clinically useful for identifying at-risk individuals). I think the paper could still include some more detail about this aspect. (See comments and response 1)

I also had a look at the other reviewer comments and the authors' responses. In general, I think that the authors have responded well, but there are a number of items which I think the authors ought to highlight in the paper itself, rather than just in the response to reviewers. Since there are quite a few of these, I'll attach my annotated .pdf document so you can take a look yourself at my highlights and comments.

Response: Thanks for your great comments. We quote the previous comments and responses, highlight your additional comments (denoted as reviewer 4) in the purple boxes, and give point-by-point responses as follows:

1. " (Reviewer 1) However, as structured, several sections of the manuscript are quite repetitive (E.g., methods and results section contain close to the same paragraph) and the manuscript as a whole could be reduced by ~25% without losing any content. Additionally, the major focus of the results section is on the associations listed above, rather than the predictive models. After reading the results section, it is not clear what is included in the models and whether those models would be clinically useful tools for identifying risk. Strongly suggest removing the redundancy and including more detail about model findings and how they could be used to inform clinical practice.

Response: Thanks for the comments. We have revised the method and results sections as suggested."

Reviewer 4

2023-07-12 19:07:41

Agree that the paper could still include more detail about how the predictive models could inform clinical practice (i.e. still after the revision), and which elements would likely be included within those models

Response: Thanks again for your great comments. We have updated our primary results in Fig 2-4 and further highlighted our primary findings and potential implications for the clinical practice. To summarize:

- We found complex association patterns and a varying predictability of several PASC conditions which may represent a challenge for managing heterogeneous PASC conditions.
- Thus, we suggest using machine learning-based predictive models and EHR databases could help in identifying patients who were at risk of developing different incident PASC conditions by managing the abovementioned complexity. (Our Python codes are available at https://github.com/calvin-zcx/pasc_phenotype/tree/master/prediction)
- Among complex association patterns, we further highlight that severe acute infections, being underweight, and having baseline conditions including cancer or cirrhosis are potentially associated with overall incident PASC in the post-acute phase
- We suggest further investigation of the association between COVID-19 treatment in adults who are at high risk for severe COVID-19 and the risk of PASC beyond the acute phase, considering the generated hypothesis that preventing severe acute infection might be associated with a lower risk of incident long-term conditions.

2. *“(Reviewer 2) 4. Several other important variables should be considered including the repeat/multiple infections, vaccination status (mentioned in the discussion, but a major limitation given data indicating that this may reduce severity of illness and PASC risk), and treatment with SARS-CoV-2 specific therapies.*

Response: Thanks for your questions. We would like to do the following clarifications. First, we didn't use vaccination status because a) the vaccine began in early December 2020 and more than half of the patients got infected before the vaccine was available.

Reviewer 4

2023-07-12 19:09:51

This should be mentioned in the paper itself (which it isn't IIRC)

Even for patients after December 2020, the number of recorded patients was limited, please see the table below.

Reviewer 4

2023-07-12 19:11:06

Again, why not include this information in the paper? And explain how the analysis would need to change if it were performed on later data where vaccination was more prevalent

We defined the fully vaccinated as two shots of mRNA vaccine (Pfizer, or Moderna) or one shot of J&J, see <https://www.cdc.gov/coronavirus/2019-ncov/vaccines/stay-up-to-date.html>. In all, the

vaccine baseline covariates before the index date only accounted for 2% of the population in the covid+ group and 5% in the covid- group, barely changing the final screened conditions considering their significant hazard ratio. We acknowledged this as a limitation in the discussion section.

	INSIGHT COVID+		INSIGHT COVID-	
	N	%	N	%
Total number	61305	100%	577174	100%
Fully vaccinated - Pre-index	811	1%	17229	3%
Partially vaccinated - Pre-index	823	1%	11753	2%
No evidence - Pre-index	59671	97%	548197	95%

Second, *Studying how the vaccine influences the risk of long Covid is a huge topic by itself and left as a future study with a more sophisticated experiment design, and also relies on the ongoing efforts of cumulating and collecting more vaccination data (e.g. registry database, which is not available yet).*”

Reviewer 4

2023-07-12 19:11:37

Again, state this more explicitly in the discussion

Response: Thanks again for your constructive comments. We have stated this explicitly in the discussion-limitation paragraph as follows, “We didn’t use COVID-19 vaccination status because the publicly available COVID-19 vaccine began in early December 2020 and nearly half of the study population got infected before any vaccine was available. Regarding the remaining half of the population who got infected after December 2020, the vaccine records collected outside the hospitals were largely missing. Studying how vaccine influences the PASC is a promising future direction as in the later cohort vaccination is more prevalent, and building the linkage to more robust vaccination data (e.g., registry database) of general patients is one of our ongoing efforts.”

3. “ (Reviewer 2) 5. *Is the discharge location of patients known? This is of particular importance in the development of conditions such as pressure ulcers, which are more likely to occur in individuals living in care facilities.*

Response: Thanks for your comments. As shown in Fig 3 (subgroup/stratified analysis by the hospitalization status), we investigated factors associated with incident PASC when we stratified patients according to their hospitalization status during their acute infection. We observed pressure ulcers were associated with ICU, >=75 ages, weight loss at baseline for hospitalized patients only, but not for the non-hospitalized patients. These findings were aligned with your proposed comments that pressure ulcers were more likely to occur in individuals living in care facilities. ”

Reviewer 4

2023-07-12 19:12:22

Why not mention this in the paper?

Response: Thanks again for your suggestions. Indeed, pressure ulcers are more likely to occur in hospitalized patients or individuals living in care facilities. We followed the suggestions from our clinician groups and several reviewers and we excluded pressure ulcers from our PASC list and updated all our analyses (see updated primary results in Fig 2-4 and supporting information in Extended Fig 1-5).

4. *“(Reviewer 2) 6. Severity of illness is clearly a driving factor for PACS risk, particularly myopathies, pressure ulcers, malnutrition, and U099/B948 codes. With the exception of the latter, however, ICU stays (particularly long stays) place individuals at increased risk for these conditions regardless of Covid status. While the authors worked to control for this with their non-infected group, the lack of granularity in the medical use variable may overlook differences in true severity of illness. For example, a Covid patient who spent a month in the ICU on a ventilator would be considered as having the same severity of illness as a non-Covid patient who spend a night in the ICU after an elective surgery. Was consideration given to using a more granular variable, such as the WHO ordinal clinical severity scale?”*

Response: Thanks for your comments. We would like to do the following clarifications. First, we use hospitalization status, ICU, and ventilation status during the 1 day prior to 16 days after their index date to capture the severity of illness during the acute phase. See a copy of top rows of table1 below. These modeling of acute severity were consistent with our previous literature¹⁻⁴ and CDC work⁵.

Reviewer 4

2023-07-12 19:14:25

Think the authors are missing the point a bit here. It's a very valid point that the duration of an ICU stay could be an important factor. The authors could at least discuss this as a limitation?

Second, we conducted a stratified analysis by stratifying patients by their severity in the acute infection phase (hospitalized or non-hospitalized) and then performing statistical analysis within each stratum. The noninfected control patients were also stratified according to the hospitalized or non-hospitalized during the acute period, to capture background associations within each subgroup population. We can see different association patterns across both two groups, as summarized in Fig 3, and Associations stratified by the acute care settings section. Third, we

acknowledge potential granularity issues of acute severity and a small number of ICU patients (a limited number of Covid patients who spent a month in the ICU) in the discussion section and we would like to explore more fine-grained severity scales, like the WHO ordinal clinical severity scale, in our future analysis as more patients were accumulated.

Reviewer 4

2023-07-12 19:16:08

IMO the author should expand more on this aspect in the discussion. At the moment it's quite brief and doesn't explicitly talk about duration of ICU / hospital stay

Response: Thanks for your great comments. We have added more discussion in the limitation paragraph as follows: “We captured the acute severity of illness by hospitalization and ICU status during their acute infection phase, consistent with the existing long COVID literature. However, these modelings of acute severity can lack granularity in the medical use variable that may overlook differences in the true severity of illness. For example, a patient who spent a month in the ICU on a ventilator should not be considered as having the same severity of illness as a patient who spent a night in the ICU after elective surgery. We would like to add more granularity to acute severity modeling by capturing the duration of ICU/hospital stay or using the WHO ordinal clinical severity scale in our future analysis as more patients accumulate.”

5. “(Reviewer 2)

9. In Extended Table 1, the HR for PACS declines with increasing number of outpatient visits prior to infection. Did the authors consider that this may be a reflection that patients with greater engagement with the healthcare system prior to infection received diagnoses before infection (either resulting in or because of greater engagement with healthcare providers); conversely, patients who had limited healthcare engagement prior to Covid infection subsequently had greater opportunity to be diagnosed with new conditions, simply because of greater engagement?

Response: *Thanks for your comments. We would like to do the following clarifications. It's possible, and thus, we tried to screen likely associations by a) adjusting for a range of baseline covariates including these baseline hospitalization utilization features, and b) further compared these associations in both infected patients versus non-infected patients. The same arguments can be applied to the non-infected patients and by requiring excess risk over this control group can further mitigate the proposed concern.”*

Reviewer 4

2023-07-12 19:19:43

Again, I would suggest that the authors include and expand on the point that the reviewer has made in the limitations section of the discussion

Response: Thanks for your suggestions. We have expanded on this point in the discussion-limitation section as follows: " The identification of the incident events can be associated with healthcare utilization behaviors: patients who had limited healthcare engagement before COVID-19 infection subsequently might have a greater opportunity to be diagnosed with new conditions simply because of less captured baseline status. Thus, in our analysis, we compared identified associations with those in non-infected patients with similar baseline characteristics including healthcare utilization behavior. In addition, we will also explore clinical notes to better capture incidence events in our future analysis. "

6. " (Reviewer 3) • To define PASC, authors use the incidence of a diagnosis from a long list of conditions (from depression and anxiety to myopathies and acute kidney failure) from 31 to 180 days after COVID infection. This will likely include a lot of false positives (people can have those conditions, regardless of COVID too). They capture all of these using ICD-10 codes. The only ICD-10 codes that are specific (though not necessarily sensitive) for PASC are U099 and perhaps B948. Authors should do a sensitivity analysis, in which they only use these two diagnoses to capture the outcome of PASC. This is important because some of the risk factors they are claiming to be related to PASC (such as obesity) are indeed risk factors for COVID-unrelated incidence of those conditions in the long list, so using a narrow, more specific definition for PASC may validate (or refute) if these risk factors are indeed related to PASC.

Response: Thanks for your comments. We would like to do the following clarifications. First, we have used the general PASC codes U099/B948 as one outcome in our PASC list and we have reported the association results regarding U099/B948 codes in our results section. As shown in Figure 1, patients with weight loss or obesity (BMI ≥ 30), or who got hospitalized or in ICU during their acute period were at a higher risk of being diagnosed with these general PASC codes. Second, the U099 codes were effective since 10/1/2021, which can not capture patients who got infected before that. Third, defining PASC is still an open question and our PASC conditions were consistent with our previous literature ¹⁻⁴, which usually exhibited excess burden over the non-infected control group. Regarding association analyses, to reduce the change of false findings, we first adopted non-infected control groups and required the adjusted hazard ratio value of the identified association estimated from the infected patients to be larger than the value estimated from non-infected patients, and we further adopted very stringent significance level corrected due to the multiple test settings.

Reviewer 4

2023-07-12 19:26:35

This should be stated in the paper

”

Response: Thanks for your suggestions. We have added the highlighted text into the result section (the severity of acute infection-subsection). “In addition, concerning being diagnosed with general PASC codes U099/B948 (the U099 code, namely unspecified post-COVID-19 condition, was effective since 10/1/2021), hospitalized or ICU patients had a 2.2-fold and 4.3-fold higher risk respectively than non-hospitalized patients.”

Reference

1. Zhang, H. *et al.* Data-driven identification of post-acute SARS-CoV-2 infection subphenotypes. *Nat. Med.* 1–10 (2022) doi:10.1038/s41591-022-02116-3.
2. Zang, C. *et al.* Data-driven analysis to understand long COVID using electronic health records from the RECOVER initiative. *Nat. Commun.* **14**, 1948 (2023).
3. Khullar, D. *et al.* Racial/Ethnic Disparities in Post-acute Sequelae of SARS-CoV-2 Infection in New York: an EHR-Based Cohort Study from the RECOVER Program. *J. Gen. Intern. Med.* (2023) doi:10.1007/s11606-022-07997-1.
4. Thaweethai, T. *et al.* Development of a Definition of Postacute Sequelae of SARS-CoV-2 Infection. *JAMA* (2023) doi:10.1001/jama.2023.8823.
5. Post-COVID-19 conditions among adult COVID-19 survivors aged 18–64 and ≥65 years — Cerner Real World Data, United States, March 2020–November 2021. <https://stacks.cdc.gov/view/cdc/117411>.

REVIEWERS' COMMENTS:

Reviewer #2 (Remarks to the Author):

The authors have addressed several of the concerns raised by the Reviewers. Globally, it remains unclear how the presented information, however, should/can be used clinically. Specifically, given the limitations of the dataset, particularly the missingness of vaccine status, which has been shown in multiple studies to have an association with PACS, the implications for use of these results is unclear. Further, while attempts were made to remove/minimize background associations, many of the highlighted associations have clear non-Covid related associations. For example, stating that older individuals are more likely to develop dementia, and that those with a BMI<18.5 are more likely to be diagnosed with malnutrition likely represents underlying patient characteristics and known disease state processes, rather than sequela of their Covid infection. An understanding of how this may improve clinical care or push the study of PASC forward is unclear.

REVIEWERS' COMMENTS:

Reviewer #2 (Remarks to the Author):

The authors have addressed several of the concerns raised by the Reviewers. Globally, it remains unclear how the presented information, however, should/can be used clinically. Specifically, given the limitations of the dataset, particularly the missingness of vaccine status, which has been shown in multiple studies to have an association with PACS, the implications for use of these results is unclear. Further, while attempts were made to remove/minimize background associations, many of the highlighted associations have clear non-Covid related associations. For example, stating that older individuals are more likely to develop dementia, and that those with a BMI < 18.5 are more likely to be diagnosed with malnutrition likely represents underlying patient characteristics and known disease state processes, rather than sequela of their Covid infection. An understanding of how this may improve clinical care or push the study of PASC forward is unclear.

Response: Thanks for your great suggestions.

First, in the Discussion-limitation paragraph, we have acknowledged this limitation, and added associated references, and discussed the vaccine issues in more detail. See below

“We acknowledge the limitation in not using COVID-19 vaccination status because the publicly available COVID-19 vaccine began in early December 2020 and nearly half of the study population got infected before any vaccine was available. Regarding the remaining half of the population who got infected after December 2020, the vaccine records collected outside the hospitals were largely missing. In addition, vaccinated patients can still develop severe infection,³⁴ which was identified as a risk factor for Long COVID by our analysis and others.³⁵ In addition, the effect of COVID-19 vaccine on Long COVID are not consistent and still need further investigation.³⁶⁻³⁸ Studying how COVID-19 vaccine influences the PASC is a promising future direction as in the later cohort vaccination is more prevalent, and building the linkage to more robust vaccination data (e.g., registry database) of general patients is one of our ongoing efforts.”

Second, regarding background associations, we have developed Criterion 3 and Criterion 4 (method – association analysis), aiming to identify the risk associations that have the portion that can be attributed to the SARS-CoV-2 infection, in addition to the background associations in non-infected patients. We conducted sensitivity analysis in results-Sensitivity section, supplementary fig 1, and we further added more discussion regarding these comments in the discussion-limitation paragraph. As reviewer mentioned, some identified associations should be interpreted with caution. For example, older individuals are more likely to develop dementia, and those with a

BMI<18.5 are more likely to be diagnosed with malnutrition likely represents underlying patient characteristics and known disease state processes. To quantify the potential exacerbation effect (if any) of SARS-CoV-2 infection on some known risk associations remains an open question. Further studies are also warranted to investigate the basic mechanisms of developing Long COVID.

Third, our study tries to uncover the complex association patterns of baseline factors and Long COVID, and varying predictability of different Long COVID conditions. We find that severe acute SARS-CoV-2 infection, being underweight, and having baseline comorbidities including cancer and cirrhosis are likely associated with increased risk of having incident Long COVID, and our developed machine learning models could be used to predict a variety of incident Long COVID conditions. We have added these implications in the abstract and discussion.